# Raised Leptin and Pappalysin2 cell-free RNAs are the hallmarks of pregnancies complicated by preeclampsia with fetal growth restriction

Sungsam Gong [1,2], Carlo Randise-Hinchliff[3], Suzanne Rohrback[3], Jing Yin Weng[4,5], Komal Singh[4], Sarah Shultzaberger [3,6], Ulla Sovio [1,2], Emma Cook [1], Fiona Kaper[3,7], Gordon C. S. Smith [1,2,7] & D. Stephen Charnock-Jones [1,2,7] ✉

Preeclampsia (PE) and fetal growth restriction (FGR) complicate 5-10% of pregnancies and are major causes of maternal and fetal morbidity and mortality. Here we demonstrate that measuring circulating cell-free RNAs (cfRNAs) from maternal plasma can accurately predict pregnancies complicated by the combination of PE and FGR. We investigated 751 maternal plasma samples from 195 pregnant women (39 cases; 156 non-cases). We developed machine learning models from our discovery cohort (15 cases; 60 non-cases) and evaluated their predictive performances internally (24 cases; 96 controls) and externally (40 cases; 73 non-cases). We found circulating leptin (*LEP*) and pappalysin2 (*PAPPA2*) cfRNAs are the strongest cfRNA predictors of complicated pregnancies, each with an area under the receiver operating characteristic curve (AUC) of ~0.82. Using an external validation dataset of women with established PE, the combination of *LEP* and *PAPPA2* had an AUC ~0.951. Our findings show that cfRNAs can predict complications of human pregnancy.

Preeclampsia (PE) is a common disease of pregnancy that affects ~7% of all pregnant women and it is a major cause of maternal and neonatal morbidity and mortality[1–3]. The most common clinical manifestation of PE is a new onset of hypertension with impairment of at least one organ, mainly the kidney with proteinuria, in the second half (i.e., after 20 weeks) of pregnancy[4]. Preeclampsia, typically the early onset (<34 weeks) form, shares common pathophysiological features with fetal growth restriction (FGR), a condition in which the fetus fails to achieve its genetically determined growth potential, which is caused by insufficient placental function[5]. In the two "placental syndromes", it is widely recognized that deficiencies in extra-villous trophoblast invasion and failure of placental vascular remodelling contribute to placental malperfusion (i.e. suboptimal blood flow) which in PE induces oxidative and endoplasmic reticulum stress and releases pro-inflammatory and antiangiogenic factors that result in maternal endothelial dysfunction, the hallmark of the condition[6]. These associations are exploited clinically by the measurement of such two factors (and their ratio), soluble fms-like tyrosine kinase1 (sFLT1) and placental growth factor (PlGF), which are now well-established biochemical assays used in clinical diagnosis of PE[7].

The identification of fetal cell-free DNA (cfDNA) circulating in maternal blood[8] greatly changed the way prenatal screening is

[1]Department of Obstetrics and Gynaecology, University of Cambridge; NIHR Cambridge Biomedical Research Centre, Cambridge, UK. [2]The Loke Centre for Trophoblast Research, Department of Physiology, Development and Neuroscience, University of Cambridge, Cambridge, UK. [3]Illumina Inc., San Diego, CA, USA. [4]Illumina Inc., Foster City, CA, USA. [5]Present address: Color Health Inc, Burlingame, CA, USA. [6]Present address: Pleno Inc, San Diego, CA, USA. [7]These authors contributed equally: Fiona Kaper, Gordon C. S. Smith, D. Stephen Charnock-Jones. ✉e-mail: dscj1@cam.ac.uk

conducted by providing more accurate and less invasive, therefore safer tests. For example, fetal chromosomal aneuploidies such as Down's syndrome (trisomy 21), Patau's syndrome (trisomy 13) and Edward's syndrome (trisomy 18) are detectable from maternal blood as early as 10 weeks of gestation, and such tests are being offered by the National Health Service in the UK or by private clinics in many countries. Other fetal abnormalities, mainly for monogenic disorders, are also detectable by sequencing a panel of genes from maternal blood to identify the presence of casual variants responsible for some inherited diseases[9]. Three years after the first finding of cfDNA, Poon et al.[10] reported the presence of fetus-derived cell-free RNA (cfRNA) circulating in maternal blood by detecting chromosome Y transcripts—a clear signal from the male fetus. Later, more studies reported that in early pregnancy, <1% of cfRNAs are of fetal origin, mainly the placenta, and the proportion increases to ~4% around 18 weeks and up to ~15% toward the end of the second trimester[11,12]. Unlike cfDNA, which conveys the fixed genetic information of the fetal genome, cfRNA could be measured quantitatively to provide a snapshot of dynamic maternal, placental and fetal repertoire of mRNA at the time it was measured. Therefore, it could be an ideal source of information providing molecular signatures that are indicative of adverse pregnancy outcomes, especially for placental diseases such as PE and FGR. However, compared to cfDNA, the use of cfRNA in prenatal screening is relatively new, and only a few studies reported the use of cfRNA in identification of complicated pregnancies, such as spontaneous preterm birth[13], PE[14-16], and FGR[17].

In this study, we show that measuring circulating cfRNAs from maternal plasma can accurately predict pregnancies complicated by PE combined with FGR. We use samples from a well-phenotyped prospective cohort of 4512 pregnant women from which samples and data were collected at 4 time points in pregnancy. This cohort has been previously described[18,19]. We selected women with pregnancies complicated by PE with FGR as this combination is a severe phenotype and likely to have the strongest signal in a cfRNA dataset. This study was based on a total of 751 maternal plasma samples from 195 pregnant women (39 cases; 156 non-cases), which were divided into the discovery cohort, where our predictive models were developed, and the validation cohort where the models were internally evaluated. For exonic enrichment we use customised probes that were tailored to the placenta transcriptome and sequenced an average of ~300 million reads per sample. We evaluate 11 machine leaning methods for selecting the best combination of predictors. Finally, we validate the predictive performance of our models using an externally-sourced RNA-seq dataset where the cohort and the sequencing data were generated independently.

## Results

### Overview of the current study

The aim of this study was to identify cell-free RNA (cfRNA) transcripts circulating in the maternal blood that are predictive of adverse pregnancy outcome and the overall workflow is shown as a schematic diagram in Fig. 1. The cases studied were pregnancies which exhibited the combination of fetal growth restriction (FGR) and preeclampsia (PE) from the Pregnancy Outcome Prediction Study[18,19]. We divided our cohort into discovery and validation groups. The discovery group consisted of 15 PE with FGR cases resulting in preterm delivery (<37 week of gestational age) and the validation group consisted of 24 PE with FGR resulting in term delivery. Each case was paired with ~4 matched controls (see Methods) hence a total of 60 and 96 healthy control samples, with no overlap, were included in the discovery and the validation group, respectively. The maternal blood was drawn at around 12, 20, 28 and 36 weeks of gestational age (wkGA) and 2 ml of plasma from each of the blood samples was used to extract RNA. Blood samples were obtained routinely at the given gestational time points as part of the research protocol and not in

response to a confirmed or suspected diagnosis of PE or FGR. A total of 751 maternal plasma samples from 195 pregnant women (39 cases; 156 non-cases) were collected and samples from a case subject at a given gestational age were analysed in the same batch as the matched controls which were also obtained at the same gestational age (Supplementary Data 1). All lab work was carried out by operators blind to case/control status. We tested several RNA-seq data processing pipelines and selected the most reliable approach based on the performance of predicting fetal sex by measuring the extent of chromosome Y (chrY) encoded transcripts (see Supplementary Information – selection of RNA-seq quantification method). At the discovery stage, cfRNAs that were differentially expressed in cases compared to controls were detected at each of the gestational age group and 11 machine learning (ML) methods were compared to find the best predictive models by controlling the number of predictor cfRNAs. The best performing model from the discovery stage was validated across each gestational age group in our internal validation dataset. We also analysed the external validation dataset[14] which reported cfRNAs in plasma samples from women with an established diagnosis of PE.

### Differentially expressed cfRNAs from the discovery dataset

After careful quality control of sequencing data, we considered a total of 15,150 eligible genes expressed in ≥10% of samples having ≥10 reads (see Methods for details). To identify differentially expressed genes (DEG), we applied multiple approaches, including a univariate logistic regression and two most widely used Bioconductor R packages, DESeq2[20] and edgeR[21], to each gestational age dataset of the discovery cohort (i.e., 12, 20, and 28wkGA). Using logistic regression, we were unable to find any significant DEG at 12 and 20wkGA, but found a total of 8,054 genes from the 28wkGA samples, which rejected the null hypothesis that the odds ratio (OR) was equal to 1 in the case and control group (adjusted $p$-value < 0.05 Benjamini-Hochberg multiple test correction[22]; see Supplementary Data 2). The histogram of the distribution of the uncorrected $p$-values suggested that there was a highly significant excess of low $p$-values ($p$-value < $2.2 \times 10^{-16}$; Kolmogorov–Smirnov test; Fig. 2A). Using edgeR[21], we could not detect significant DEGs at 12 and 20wkGA, but identified a total of 5898 genes that were significantly differently expressed (i.e., adjusted $p$-value < 0.05; Benjamini-Hochberg multiple test correction method) from the 28wkGA samples (Fig. 2B; Supplementary Data 3). Similarly, DESeq2[20] was also unable to identify any significant DEG from the 12 and 20wkGA groups but identified 1445 DEGs and these were a subset of the 5898 genes identified using edgeR (Supplementary Data 4). Out of 5898 DEGs identified by edgeR, 5499 (93%) were included in the list of 8054 genes from the univariate logistic regression method. Having identified differentially expressed genes from the 28wkGA samples, we selected the top 1% genes by the $p$-values based on DESeq2 and edgeR, and AUC (Area Under the receiver operating characteristic Curve), and the AIC (Akaike Information Criterion) from univariate logistic regression. 345 genes (see Supplementary Data 5) were in the top 1% of one or more of the four methods and a Venn diagram demonstrated that 17 genes were identified by all four selection methods (Fig. 2C). Among these 17 shared genes (Table 1), the transcript abundance levels of *LEP* and *PAPPA2* cfRNAs were positively related with the case (i.e. higher in cases), whereas the other 15 cfRNAs were negatively related (i.e., lower in cases). Overall, as shown from the heatmap (Fig. 2D), the disease outcomes of 72 samples were separated by the cfRNA levels of the 17 shared genes.

### Machine learning approach to find the best feature selection method

Having identified the 17 shared differentially expressed genes from the 28wkGA samples of the discovery dataset, we proceeded to explore various ML methods to find the best approach in terms of

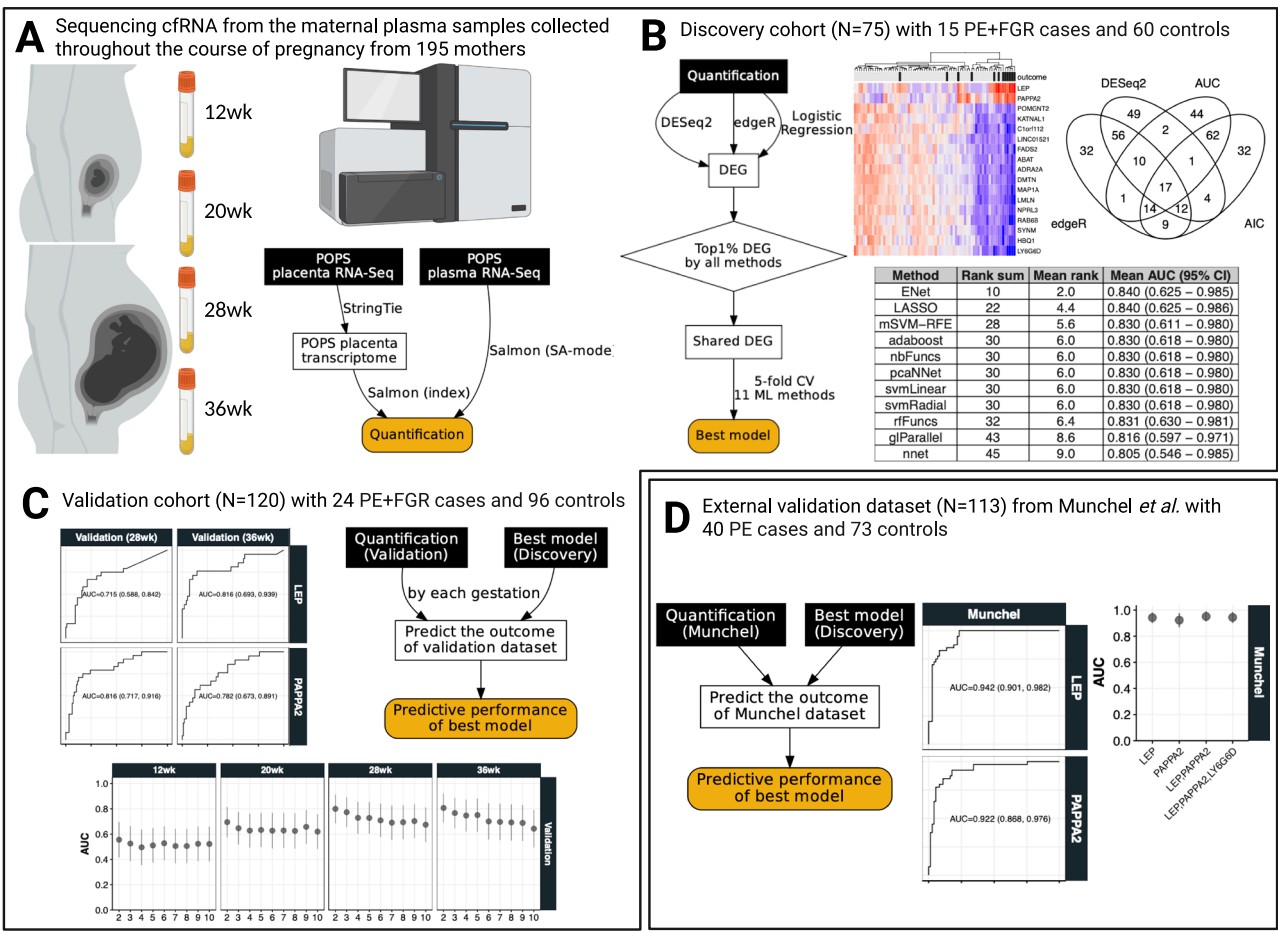

**Fig. 1 | Schematic diagrams showing the current study design. A** cfRNAs were sequenced from maternal plasma samples taken at 12-, 20-, 28- and 36wkGA from a total of 195 pregnant women, and they were quantified using a carefully selected method based on a benchmark of detecting the chromosome Y transcripts from female pregnancies (see Supplementary Information for additional detsail). **B** in the discovery stage, a set of differentially expressed genes (DEGs) were detected using various methods and we selected the best performing predictive model out of 11 machine learning (ML) methods. Performance was assessed by cross-validated AUC (Area Under the receiver operating characteristic Curve based on 5-fold cross validation with 5 repetitions). **C** Having trained the best performing ML method with 2–10 cfRNA predictors in the discovery stage, their predictive performances were measured at each of gestational age using our internal but separate validation cohort. **D** Finally, the predictive performance was assessed externally using the dataset from Munchel et al. who analysed cfRNAs from the maternal plasma samples of their own cohort. **A**–**D**, in the workflow diagram, the input is shown as a rectangular box with black background and the output is shown as a rounded box in yellow background. In (**C**, **D**) the mean AUC and 95% CI are shown. Created in BioRender https://BioRender.com/bwfj19q.

predicting the outcomes given the expression profiles of the 17 shared genes. Many methods have been described to generate predictive models by selecting the best combination of predictors (predictors are often referred to as features in the machine learning literature) from a large number of potential candidates which were associated with the given disease. However, many studies simply present the result of a single method without justifying the use of that method over alternative approaches. We therefore employed a modified version of the discrete super learner approach in which the methods to be compared are selected and the measure to be employed for comparing models is identified[23]. We considered a total of 11 ML methods selected from a range of different ML approaches (Table 2) and compared their predictive performances using the AUC based on 5-fold cross-validation (CV) with 5 repetitions (Supplementary Data 6; see Methods for details). To account for different number of predictors in each model, we controlled the number of selected cfRNAs in the predictive models from 2 to 6 cfRNAs. For example, for a given number of desired cfRNAs in the model, we calculated the mean AUC from the 25 cross-validated AUCs, i.e., 5-fold CV repeated 5 times, for a given ML method, and compared the mean AUC (and their rank) across the 11 ML methods. This procedure was repeated for a selection of 2 to 6 cfRNAs in the

predictive model and the mean AUCs and the sum of their ranks were compared across the 11 ML methods. Finally, we ranked each method and identified the best method, i.e., the one with the lowest sum of the ranks across 2 to 6 cfRNAs in the model. We found that, among the 11 ML methods, regularised (or penalised) regression methods such as Elastic net[24] and LASSO (Least Absolute Shrinkage and Selection Operator[25]) yielded the highest mean AUC (0.840 for both), although Elastic net performed fractionally better than LASSO in terms of the rank sum (Fig. 3A and Supplementary Data 7). During 5-fold CV, we tuned the parameters of two Elastic net methods (ENet1 and ENet2) with different approaches (see Methods), but both returned the same results in terms of the mean AUC and the rank. These were the best among the 11 ML methods and we present the data from ENet2 (Fig. 3A and Supplementary Data 7). A feed-forward neural network-based method using R nnet (v7.3.16) package[26] showed the least predictive performance (the mean cross-validated AUC = 0.8054) and ranked bottom, followed by glParallel[27], a brute-force exhaustive search method testing all combinations of features in the regression model, and a random forest method (see Methods for details). Based on the benchmark of 11 ML methods, we chose Elastic net and trained our prediction model using the entire training dataset.

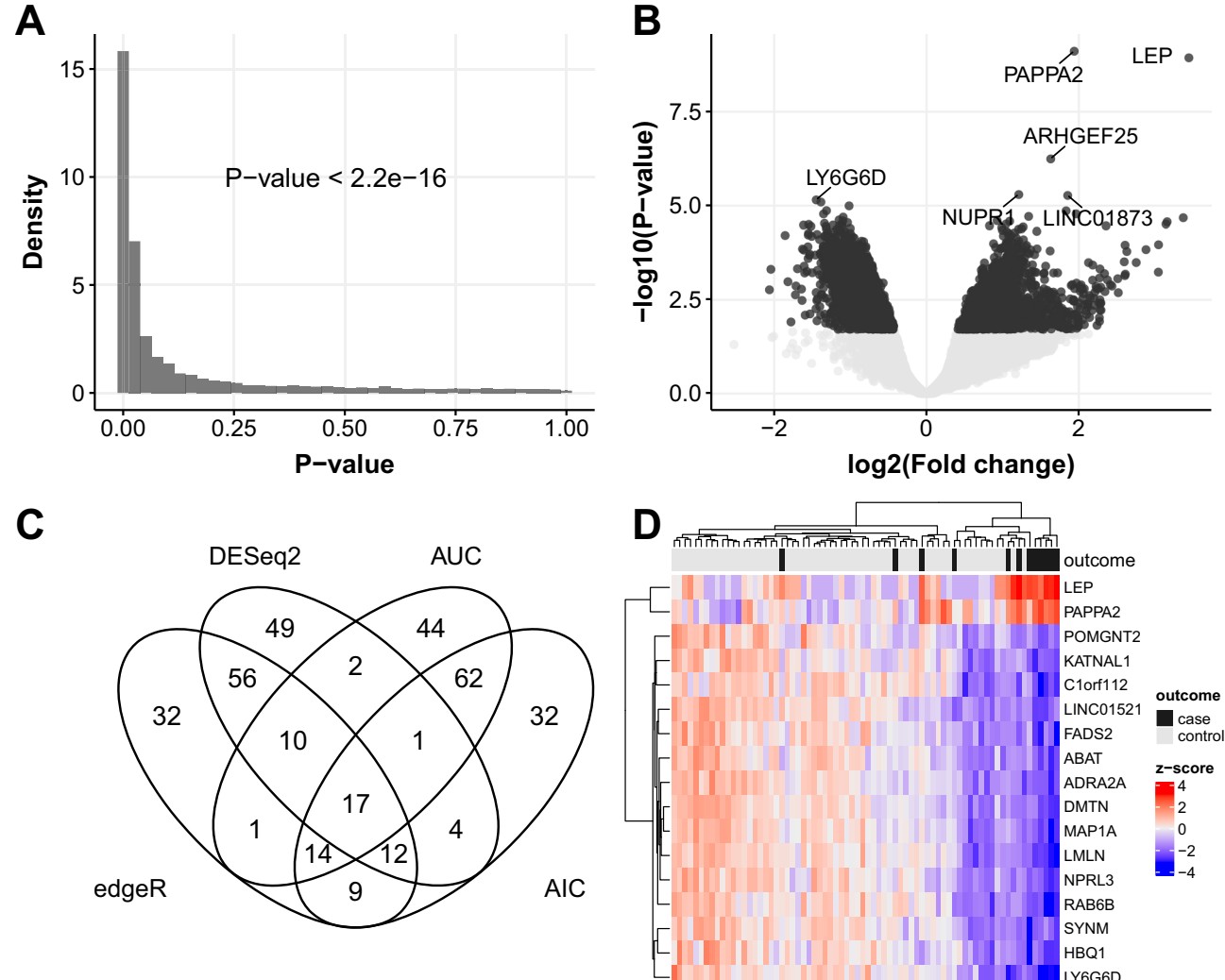

**Fig. 2 | Differentially expressed cfRNAs from the 28wkGA group of the discovery cohort. A** The histogram of the *p*-values from the univariable logistic regression of individual cfRNA models. The *p*-value was obtained from the Kolmogorov–Smirnov test (2-sided). **B** The volcano plot showing the degree of statistical significance (*p*-value) against the effect size (fold change) of the expression difference between the case and the control. The original *p*-values on the y-axis and the fold changes on the x-axis were obtained from edgeR (F-test). The

6 cfRNAs having the lowest *p*-value were indicated. After applying Benjamini-Hochberg correction method for multiple tests, those with ≤ 0.05 were indicated in dark-coloured dots, otherwise in pale-coloured dots. **C** The Venn diagram showing the number cfRNAs that overlap among the four selection criteria. **D** The heatmap showing the relative level of the 17 shared RNAs identified in the Venn diagram. The z-score represents the normalised Count Per Million (on log2-scale) values.

## Training models and best subset selection

Having identified the best performing feature selection ML method, i.e., Elastic net[24], we tuned two hyperparameters, alpha (i.e., regularization penalty) and lambda (i.e., shrinkage penalty), that lead to the number of predictors being reduced to 2 to 10 cfRNAs of the shared 17 genes. Then we trained the final models using all the 28wkGA samples of the discovery dataset in a multivariable logistic regression setting. The aim was to identify the best performing predictive model with a smaller subset cfRNAs that can be clinically useful biomarkers. We found Elastic net selected cfRNAs incrementally as the number of predictor cfRNAs increased in the model. For example, *LY6G6D* was added to the two cfRNA model (*LEP + PAPPA2*) and *HBQ1* was added to the three cfRNA model (*LEP + PAPPA2 + LY6G6D*) (Supplementary Data 8). As in a univariate logistic regression setting, we accessed the performance of multivariate models using the AUC and information criteria. In training, we found that the AUC of multivariate model increased as the number of predictor cfRNAs was increased from 2 to 10 (Fig. 3B). However, when we corrected the overestimation bias (also known as optimism) using leave pair out cross validation (LPOCV)[28],

the opposite trend was observed: the LPOCV corrected AUC (LPOCV-AUC) decreased as the number of predictor cfRNA increased, indicating likely overfitting of the model with increasing numbers of predictors. This interpretation was supported by an increasing trend of the information criteria both from the Akaike information criterion (AIC) and the Bayesian information criterion (BIC), where lower values suggest relatively better models. The LPOCV-AUC was highest (0.8875) in the two cfRNA model (*LEP + PAPPA2*) followed by the four cfRNA model (*HBQ1 + LEP + LY6G6D + PAPPA* with an LPOCV-AUC of 0.8778). Detailed code describing this is available in GitHub[29].

## Internal and external validation of predictive performance

Using the final 2- to 10-cfRNA models trained from the 28wkGA samples of the discovery dataset, we assessed their prediction performances using the internal validation dataset across the four gestational time points (i.e., 12, 20, 28 and 36wkGA). Overall, the predictive performance of the cfRNA models increased as the gestational age of the validation samples progressed, and they worked best in the 36wkGA samples of the validation cohort (mean AUC = 0.722 across 2- to 10-

**Table 1 | The 17 shared differentially expressed genes identified from the discovery cohort at 28wkGA**

| Gene | AUC (95% CI) | Odds Ratio (95% CI) | *p*-value[§] | Fold Change[+] | CPM[*] |
|---|---|---|---|---|---|
| *PAPPA2* | 0.901 (0.816–0.986) | 5.029 (2.417–13.388) | 1.5e-04 | 2.9 | 2.0 |
| *HBQ1* | 0.861 (0.750–0.972) | 0.258 (0.111–0.493) | 2.7e-04 | 0.4 | 16.3 |
| *LEP* | 0.851 (0.675–1.000) | 3.675 (2.044–8.093) | 1.4e-04 | 3.6 | 0.2 |
| *ABAT* | 0.849 (0.746–0.952) | 0.278 (0.122–0.536) | 5.5e-04 | 0.4 | 66.7 |
| *NPRL3* | 0.846 (0.726–0.966) | 0.278 (0.122–0.531) | 4.8e-04 | 0.5 | 175.3 |
| *LMLN* | 0.844 (0.733–0.956) | 0.346 (0.182–0.593) | 3.4e-04 | 0.4 | 16.3 |
| *MAP1A* | 0.844 (0.736–0.952) | 0.329 (0.163–0.586) | 5.1e-04 | 0.4 | 351.9 |
| *POMGNT2* | 0.842 (0.716–0.968) | 0.239 (0.091–0.511) | 9.4e-04 | 0.5 | 7.4 |
| *RAB6B* | 0.838 (0.698–0.977) | 0.314 (0.152–0.553) | 3.0e-04 | 0.4 | 170.1 |
| *ADRA2A* | 0.836 (0.713–0.959) | 0.307 (0.144–0.569) | 5.8e-04 | 0.4 | 11.7 |
| *C1orf112* | 0.836 (0.710–0.962) | 0.348 (0.182–0.597) | 3.8e-04 | 0.5 | 13.4 |
| *KATNAL1* | 0.836 (0.710–0.963) | 0.306 (0.142–0.571) | 6.5e-04 | 0.4 | 39.4 |
| *LINC01521* | 0.832 (0.677–0.987) | 0.276 (0.117–0.536) | 6.9e-04 | 0.5 | 1.1 |
| *SYNM* | 0.832 (0.710–0.954) | 0.328 (0.157–0.590) | 8.0e-04 | 0.4 | 170.0 |
| *LY6G6D* | 0.831 (0.688–0.973) | 0.339 (0.177–0.565) | 2.0e-04 | 0.4 | 1.5 |
| *FADS2* | 0.828 (0.708–0.948) | 0.291 (0.129–0.552) | 6.8e-04 | 0.5 | 39.3 |
| *DMTN* | 0.825 (0.701–0.949) | 0.321 (0.155–0.586) | 6.6e-04 | 0.5 | 789.1 |

[§]from logistic regression, [+]from DESeq2 (Wald test), [*]Count Per Million

**Table 2 | 11 machine learning methods used in this study**

| Category | Abbreviation | Comment and Source | Reference |
|---|---|---|---|
| Elastic net | ENet | Parameter α and tuned via caret R package and parameter λ via glmnet R package. | 24,55,56 |
| Least Absolute Shrinkage and Selection Operator | LASSO | Parameter λ tuned via caret R package. | 24,55,56 |
| Support Vector Machine | mSVM-RFE | Multiple support vector machine with recursive feature elimination by Dual et al. 2005. | 53 |
| | svmLinear | Support vector machines with linear kernel via kernlab R package. | 57 |
| | svmRadial | Support vector machines with radial basis function kernel via kernlab R package. | |
| Neural Network | pcaNNet | Neural networks with feature extraction via nnet R package. | 26,55 |
| | nnet | Feed forward neural networks and multinomial log-linear models via nnet R package. | |
| Naïve Bayes | nbFuncs | Naïve Bayes classifier via klaR R package. | 58 |
| AdaBoost | adaboost | AdaBoost classification trees via fastAdaboost R package. | 59 |
| Random Forest | rfFuncs | Fandom forest via randomForest R package. | 60 |
| Brute-force Search | glParallel | A brute-force exhaustive search method testing all combinations of features in the regression model (https://gitlab.developers.cam.ac.uk/ssg29/glparallel). | 27 |

cfRNA models) followed by 28wkGA (mean AUC = 0.709) and 20wkGA (mean AUC = 0.629) (Fig. 4A). However, for the 12wkGA samples, the mean AUC was only 0.521 across the 2- to 10-cfRNA models. Interestingly, we found that increasing the number of predictor cfRNAs did not improve the predictive performance of the model, which is in-line with the LPOCV-AUC and the information criteria shown in Fig. 3B. The AUC gradually deteriorated when the number of cfRNAs increased, which was especially evident in the 28wkGA and 36wkGA validation samples (Fig. 4A). The two-cfRNA model (*LEP + PAPPA2*) performed best among the 2- to 10-cfRNA models in each gestational age of the validation cohort (AUC = 0.807 at 36wkGA; AUC = 0.800 at 28wkGA; AUC = 0.695 at 20wkGA; AUC = 0.556 at 12wkGA).

In addition to the multivariable models, we next checked the predictive performance of two univariable cfRNA predictors, *LEP* and *PAPPA2*, that performed best from the 28wkGA discovery cohort based on the AIC and AUC score, respectively. Surprisingly, *PAPPA2* as a sole predictor outperformed the best two cfRNA (*LEP + PAPPA2*) at each of the 12wk, 20wkGA and 28wkGA gestational ages in the validation cohort (Fig. 4B), with an AUC reaching its highest at 28wkGA (AUC = 0.816, 95% CI = 0.717–0.916). A similar pattern was seen in the 20wkGA

samples in the discovery cohort (i.e., the model being trained using the 28wkGA sample in the discovery cohort and tested in the 20wkGA from the same cohort) with its AUC 0.812 (95% CI = 0.707–0.917). This is a very close match to the performance in the 28wkGA samples of the validation cohort. The *LEP* model, however, did not work well for the 12wk, 20wkGA and 28wkGA gestational age groups of the validation cohort compared to the two and three cfRNA models, but it outperformed all other models in the 36wkGA validation samples (AUC = 0.816, 95% CI = 0.693–0.939, Fig. 4B).

Finally, we tested our predictive models using a publicly available RNA-seq dataset from Munchel et al.[14]. In contrast to the POP study where samples were obtained routinely, the Munchel et al. study used blood samples from mothers diagnosed with PE. This external validation dataset consists of 40 PE cases with preterm birth and 73 healthy controls, with the mean gestational age when blood was collected was 30.5wkGA. The AUCs of all the three models considered (i.e. *LEP*, *PAPPA2*, and *LEP + PAPPA2*), surpassed 0.922 with very narrow confidence intervals (Fig. 4C and Fig. 4D). The predictive performance of the two-cfRNA model (*LEP + PAPPA2*) was the highest (AUC = 0.951, 95% CI = 0.912-0.990) followed by *LEP* as a sole predictor (AUC = 0.942,

**A**

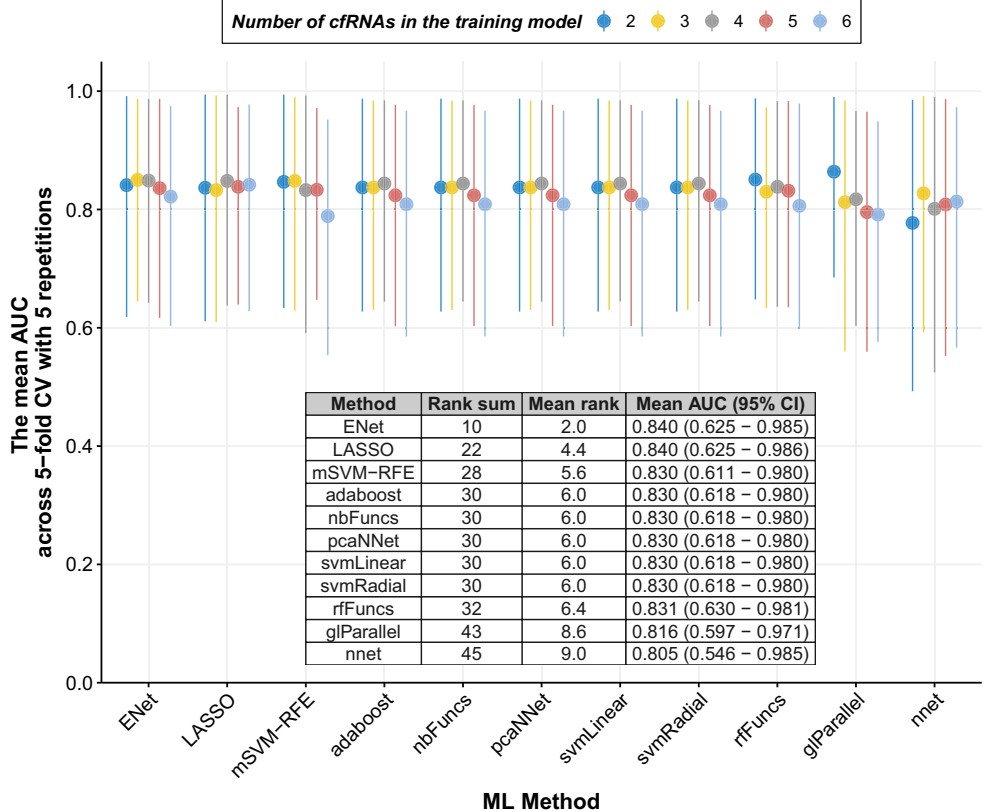

**B**

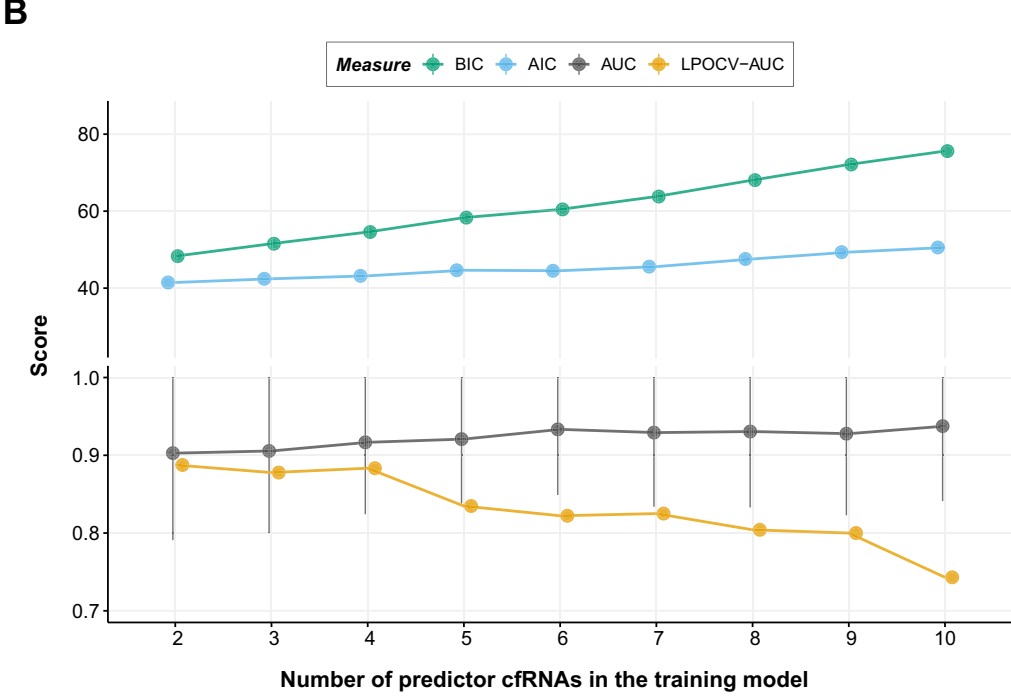

**Fig. 3 | The summary of 5-fold CV and the final training models. A** The summary of 5-fold CV with 5 repetitions for 11 ML methods. For a given ML method, the filled circle represents the mean AUC across 25 tests (i.e., 5-fold with 5 repetitions) and the lines represent the 95% confidence intervals (CI). The inset table is the final summary of the 11 ML methods across 2 to 6 predictor cfRNAs as shown in the graph. **B** The final training models with 2–10 predictor cfRNAs selected by Elastic net. The models were trained using the 28wkGA samples of the discovery cohort and the AUC is from the 28wkGA samples of the discovery cohort where the model was trained. Therefore, the AUC in (**B**) is not cross validated. For the AUC, mean and 95% CI are shown. (AIC Akaike Information Criterion, BIC Bayesian Information Criterion, AUC Area Under the receiver operating characteristic Curve, LPOCV-AUC Leave Pair Out Cross Validated AUC).

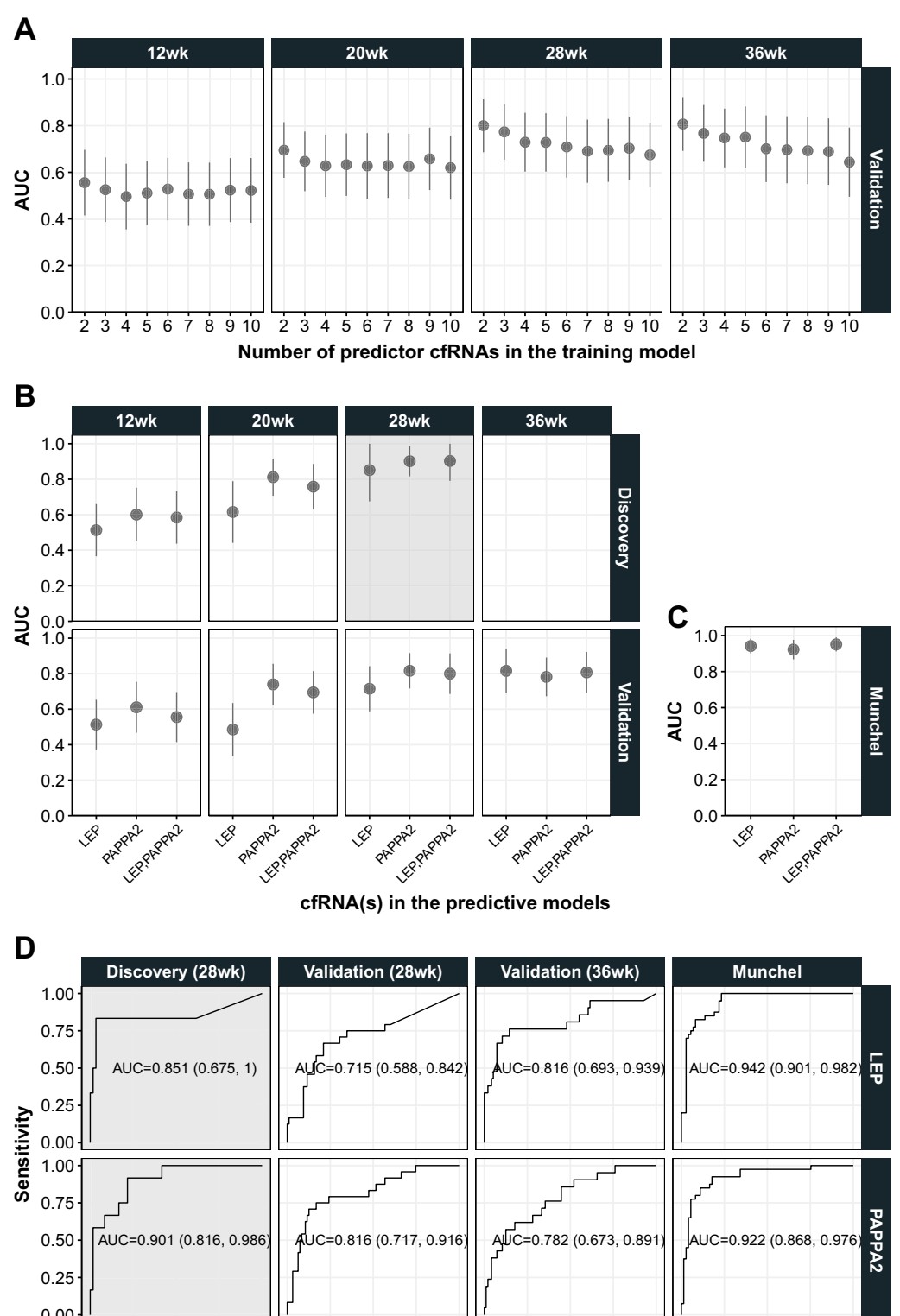

95% CI = 0.901–0.982), and *PAPPA2* as a sole predictor (AUC = 0.922, 95% CI = 0.868-0.976). We evaluated models generated using all the 28wkGA samples from both discovery (preterm delivery) and validation cohort (term delivery) cohorts. The same two-cfRNAs (*LEP + PAPPA2*) were chosen by Elastic net in this combined model. However, the predictive performance of this model was slightly lower in the external Munchel dataset (AUC = 0.944, 95% CI = 0.901–0.988) than

that of the original model trained in discovery cohort only (Supplementary Data 9 and Supplementary Fig. 2).

**Abundance of LEP and PAPPA2 cfRNA by gestational age**
In early gestation of pregnancy (12wk), the transcript abundance of *LEP* and *PAPPA2* was not significantly different between the case and the control groups (Fig. 5A). In the mid gestation samples (20wkGA), the

**Fig. 4 | The predictive performance of various cfRNA models using the internal and external validation datasets. A** The AUC scores of the 2- to 10-cfRNA models, chosen by Elastic net, using the 12wk, 20wk, 28wkGA and 36wkGA samples from the internal validation cohort. **B** the AUC scores of two univariable cfRNA models (*LEP* and *PAPPA2* separately) and the best performing multivariable cfRNA models (*LEP* + *PAPPA2*) using the discovery and the validation datasets. The best performing multivariable model, i.e., two cfRNA model (*LEP* + *PAPPA2*), was chosen by Elastic net as shown in (**A**). **C** The AUC scores of three selected cfRNA models validated on the external dataset (14). **D** Individual Receiver Operating

Characteristic (ROC) curves of two univariable *LEP* and *PAPPA2* cfRNA models using the discovery, the validation cohort, and the external dataset (14). In (**B, D**), the panel coloured in grey background indicates the training dataset (i.e., the 28wkGA samples of the discovery cohort) where the corresponding models were fitted. Therefore, the AUC is from the same dataset where the model was trained, i.e., not cross validated. In Supplementary Fig. 3, the z-scores of *LEP* and *PAPPA2* are plotted by the gestational age and the type of datasets (i.e., internal and external validation dataset) as the same format in (**B**). In (**A–C**) the mean AUC and 95% CI are shown.

abundance of *PAPPA2* cfRNA was significantly higher in the PE cases compared to the controls, but there was no significant difference in *LEP* cfRNA. From the third trimester (≥28wkGA), both *LEP* and *PAPPA*2 transcripts were significantly higher ($p$-value< 0.01, Mann-Whitney test) in the PE cases compared to the controls and the magnitude of their differences was greater at advanced gestational ages, e.g. the 36wkGA. Interestingly, when the abundance of *LEP* cfRNA was plotted along the line of gestational age, it followed a U-shape, i.e., higher in early (12wkGA) and late (36wk) pregnancy and lower in mid-gestation (20wkGA and 28wkGA). Specifically, based on the validation cohort, the abundance of LEP transcript decreased almost two-fold from the 12wkGA to 20wkGA ($p$-value = $5.4 \times 10^{-26}$, Mann-Whitney test), but it increased -1.8 fold from the 28wkGA to 36wkGA ($p$-value = $6.1 \times 10^{-18}$, Mann-Whitney test), while it remained low between 20wkGA and 28wkGA ($p$-value = 0.13, Mann-Whitney test). The same pattern was observed in the discovery cohort, i.e., two-fold decrease from the 12wkGA to 20wkGA ($p$-value = $6.2 \times 10^{-13}$) and no change between 20wkGA to 28wkGA ($p$-value = 0.22). In contrast, the abundance of *PAPPA2* cfRNA gradually increased as gestation progressed. In the discovery cohort, the Pearson's correlation coefficient between *LEP* and *PAPPA2* cfRNAs was higher at 28wk ($\rho$ = 0.63) than that at 20wk ($\rho$ = 0.56). In the validation cohort, the correlations were even weaker (0.59 and 0.32 at 28wk and 20wk respectively).

### Longitudinal analysis of cfRNAs in healthy pregnancy

Having identified gestational age-related changes in circulating *LEP* and *PAPPA2* cfRNAs (Fig. 5A), we sought to identify all cfRNAs that change in abundance as gestation progresses. For this analysis, we used a total of 96 healthy samples from the validation cohort (i.e. term delivery) and carried out two types of analyses: 1) to identify cfRNAs that differ between specific gestational ages: (i.e. 12wk to 20wk, 20wk to 28wk, and 28wk to 36wk), and 2) a longitudinal study of cfRNAs across the four gestational ages. We found a total of 129 genes for which the cfRNA levels changed significantly (76 from 12wk to 20wk; 64 from 20wk to 28wk; 55 from 28wk to 36wk, see Supplementary Data 10). *LEP* and *PAPPA2* were among these 129 genes, where *LEP* cfRNA decreased from 12wk to 20wk (adjusted $p$-value = $1.2 \times 10^{-30}$, Benjamini-Hochberg method) and *PAPPA2* cfRNA increased between 20wk to 28wk (adjusted $p$-value = $6.9 \times 10^{-5}$) and 28wk to 36wk (adjusted $p$-value = $2.6 \times 10^{-39}$). Of the 129 genes, 16 (*ALPP, C2orf72, CAPN6, CSHL1, CYP11A1, HSD3B1, LINC00967, MUC15, PAPPA, RAB3B, SVEP1, TEAD3, TFAP2A, TINCR, TMEM54,* and *VGLL3*) significantly increased between all intervals (Fig. 5B). For the longitudinal analysis, we considered a random intercept model (i.e. allowing different baseline levels for each participant) with both a linear and quadratic term of gestational age as fixed effects (see Methods). When the fitness of mixed effect model was ranked by the coefficient of determination (i.e., $r^2$), *CSHL1* showed the highest marginal $r^2$ (0.689), followed by *CAPN6* (0.688) and *RAB3B* (0.631) – see Supplementary Data 11. All the 16 aforementioned genes were highly ranked – 10 of them (*CSHL1, CAPN6, RAB3B, SVEP1, VGLL3, ALPP, LINC00967, PAPPA, HSD3B1,* and *MUC15*) were within the top 12 (Supplementary Data 11 and Supplementary Fig. 4). When the models were assessed by the significance of fixed effects, *LEP* was the most significant (adjusted $p$-value = $9.7 \times 10^{-91}$, Benjamini-Hochberg method), followed by *CSHL1* (adjusted $p$-value = $2.5 \times 10^{-44}$) and

*PAPPA2* (adjusted $p$-value = $4.53 \times 10^{-32}$) (Supplementary Fig. 5). Indeed, *LEP* showed the highest absolute coefficient values of fixed effects when the gestational age was considered as its linear (−5.62) and quadratic (1.08) term in the mixed effect model. Finally, when only the linear term was considered in the model (i.e., no quadratic term), *TTPAL* was identified as the most significantly decreasing cfRNA (adjusted $p$-value = $8.5 \times 10^{-59}$) by the gestational age, followed by *KIAA1147* (adjusted $p$-value = $2.7 \times 10^{-50}$) and *RPL37AP1* (adjusted $p$-value = $5.4 \times 10^{-41}$) (Supplementary Data 12 and Supplementary Fig. 6).

## Discussion

The main finding of the current study is that *LEP* and *PAPPA2* are the two key cfRNAs in maternal blood that are predictive of pregnancies complicated by the combination of PE and FGR. Importantly, their predictive performance is not only reproduced in our internal validation dataset but also in the external dataset from Munchel et al.[14], where the AUC of our combined *LEP* and *PAPPA2* model was 0.951 (95% CI = 0.912–0.990), and *LEP* and *PAPPA2* as sole predictors had AUCs of 0.942 (95% CI = 0.901–0.982) and 0.922 (95% CI = 0.868–0.976), respectively. This represents exceptionally accurate diagnostic performance. A surprising and atypical finding in the present study is that the predictive performance was higher in the external validation sample than in the samples where the predictive models were derived. This likely reflects the fact that the POP study samples were obtained routinely at scheduled time points in the pregnancy and the majority will have preceded the clinical manifestation of disease. In contrast, the Munchel et al. samples were obtained from women with a clinical diagnosis of preterm preeclampsia. It is plausible that true associations will become stronger with clinical progression of disease.

The sFLT1:PlGF ratio in maternal serum is used as a clinical diagnostic marker for PE. We have assessed this is the POPS cohort and the AUC was 0.95 (95% CI = 0.90–1.0 at 28 weeks gestation and 0.92–0.99 at 36 weeks gestation)[30]. Both the cfRNA and the protein tests were strongly associated with PE with FGR but the protein test appeared to perform better within sample (i.e., without testing in a separate validation cohort). Notably, the cfRNA model yielded an AUC of 0.95 in a geographically and clinically divergent cohort from the study used to generate the model; the equivalent protein data was not available for comparison.

The molecular and cellular mechanisms of release of placental and fetal cfRNAs into the maternal blood remains to be fully elucidated. However, it is clear that *PAPPA2* (also known as Pregnancy-Associated Plasma Protein A2) cfRNA is highly enriched and almost exclusively expressed in the placenta[31]. *LEP* cfRNA is also highly enriched in the placenta, but breast and adipose tissues respectively express ~2 and -3 times more than the placenta, suggesting both maternal and placental contributions in circulation. Interestingly, we and others[32–34] showed that both *PAPPA2* and *LEP* are significantly elevated in the placenta from pregnancies complicated by PE and FGR compared to healthy controls, indicating the placenta as a plausible source for the elevated levels of the two cfRNAs in maternal blood. Plasma cfRNA is partially composed of cfRNA in exosomes and extracellular microvesicles, which are known to be released by the placenta into the maternal circulation. While the perspective of the current analysis has focused on using cfRNA levels to predict and to diagnose disease, it is also

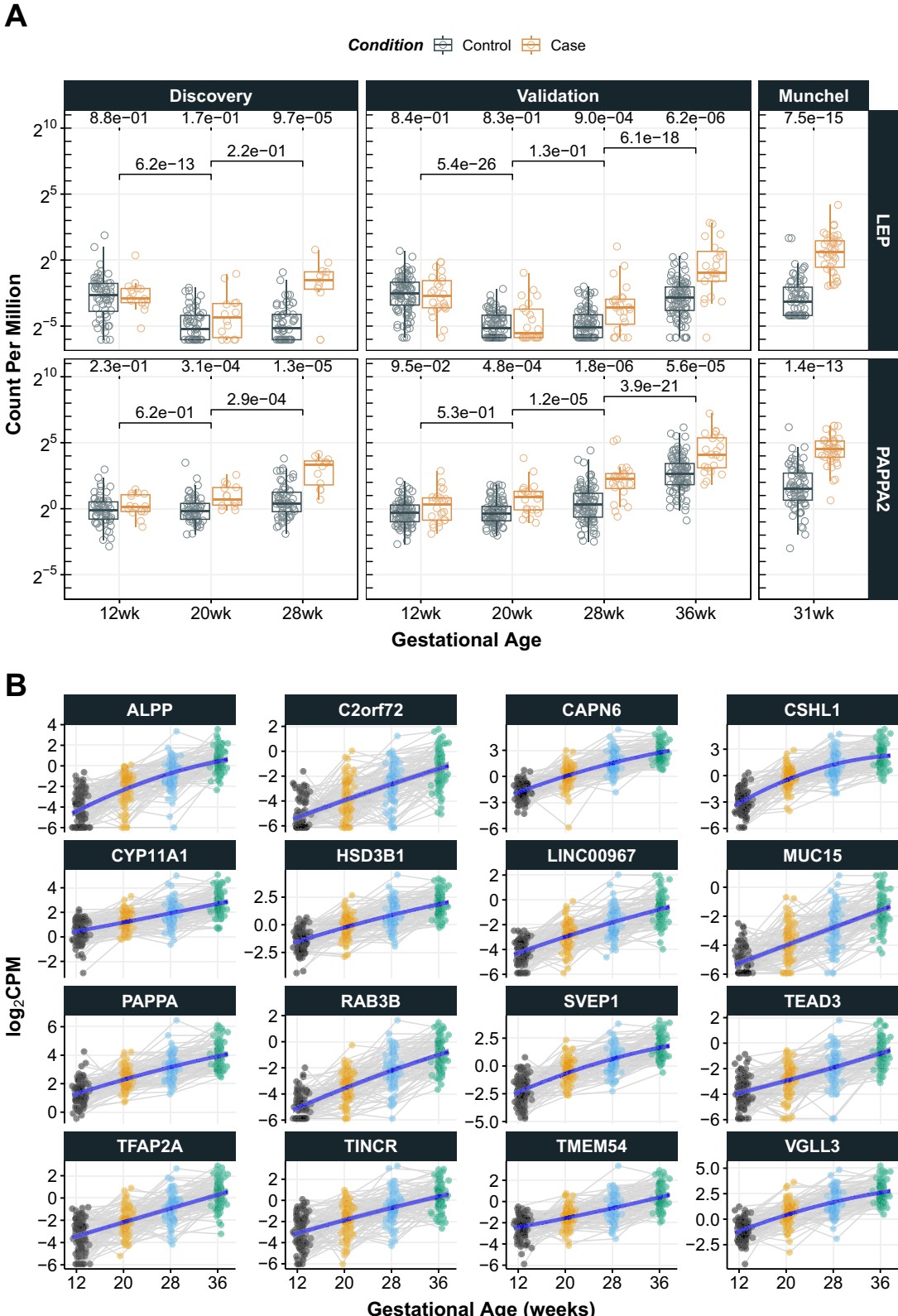

possible that the very strong association between *LEP* and *PAPPA2* cfRNA in the maternal blood is part of the pathophysiology of the disease, with the placenta controlling maternal physiology though parcelling these cfRNAs in exosomes and extracellular microvesicles which subsequently alter maternal protein production and contribute to disease pathogenesis. Pregnancy requires considerable adaptation of maternal physiology and leptin plays a major role in the regulation

of energy homeostasis, but also has endocrine functions. The function of *PAPPA2* is much less clear but the rare patients with *PAPPA2* deficiency show reduced growth and are insulin resistant[35]. Exosomal signalling from the placenta could play a role in modulating these important regulatory pathways.

Several previous studies reported circulating cfRNAs indicative of PE in maternal plasma samples[14,15,36]. Rasmussen et al.[36] reported a total

**Fig. 5 | Gestational age-related changes in cfRNA. A** The abundance levels of *LEP* and *PAPPA2* cfRNA. The count per million (CPM) on the y-axis was calculated by the edgeR Bioconductor package using the cpm function. The *p*-values shown at the top of each panel represent the statistical significance levels of cfRNA difference between the case and control samples at a given gestation age, and those shown in the middle represent the statistical significance levels between two corresponding gestational ages. All the *p*-values were calculated by the Mann-Whitney test (two sided) using the wilcox.test function of R with measurements from individual study participants. The boxes show the median and interquartile range (IQR). The vertical lines (whiskers) extended from the box represent a range of 1.5 x IQR from both ends. **B** 16 cfRNAs that increase significantly across all three gestational intervals plotted at all time points. Individual samples from each participant of the same gestational age group were coloured the same and those from the same participant were indicated in grey lines linked for each interval. The blue lines represent fitted (i.e., predicted) cfRNA abundance from the mixed effect models with two fixed terms (a linear and quadratic term of gestational age) and a random slope.

of 7 genes (*CLDN7*, *PAPPA2*, *SNORD14A*, *PLEKHH1*, *MAGEA10*, *TLE6* and *FABP1*) and the AUC of their predictive model was 0.82 (95% CI = 0.76–0.88) based on a total of 524 blood samples (72 cases and 452 non-cases) drawn at 14.5wkGA (standard deviation: 4.5wkGA). Three of these (*CLDN7*, *PAPPA2*, and *PLEKHH1*) were identified as differentially expressed genes in our 28wkGA discovery dataset (Supplementary Data 2–4). The study from Moufarrej et al.[15] reported a panel of 18 genes predictive of PE risk based on a total of 85 blood samples (24 cases and 61 non-cases) measured between 5 and 16wkGA. In their external validation, the AUC of their predictive model was as high as 0.74 (90% CI = 0.73–0.75 based on Del Vecchio dataset). 10 of these (*DERA*, *FAM46A*, *KIAA1109*, *MYLIP*, *NDUFV3*, *PRTFDC1*, *PYGO2*, *RNF149*, *TFIP11*, and *TRIM21*) were identified as differentially expressed genes in our 28wkGA discovery dataset (Supplementary Data 2–4), but none was included in our 17 shared genes list (Table 1), nor were they in the top 1% genes of genes identified by any of the four methods used (a total of 345 genes shown in Fig. 2C and Supplementary Data 5). Munchel et al.[14], which we used as external validation dataset, reported a total of 30 refined transcripts with altered abundance in 113 plasma samples (40 cases and 73 non-cases) collected at ~31wkGA. Out of their 30 reported genes, 22 were also in our DEG list, including both *LEP* and *PAPPA2*. Interestingly, we noted that the mapped read count of *LEP* and *PAPPA2* from Moufarrej et al.[15] was extremely low – the mean count was 0.46 for *LEP* and 0.99 for *PAPPA2*. In contrast, the average mapped read counts from Munchel dataset[14] were 33.8 for *LEP* and 490.5 for *PAPPA2*. In our discovery dataset, the average mapped read counts for both transcripts were comparable to the levels reported by Munchel et al. and far higher than those reported by Moufarrej et al.: 29 (12wkGA), 5 (20wkGA), 12 (28wkGA) for *LEP*, and 144 (12wkGA), 164 (20wkGA) 373 (28wkGA) for *PAPPA2* (see also Fig. 5). We could not attempt to validate our findings using the Rasmussen dataset[36] as their RNA-seq data is not in public domain.

Clearly there are a number of inconsistencies among different maternal cfRNA studies, and they could be attributed to a number of factors from the study design, laboratory experimental protocol, to the choice of analytic methodologies. However, it is very challenging to control these at each level and there is a trade-off between the choice of different methods. For example, ribosomal RNAs (rRNAs) account more than 95% of the RNAs in the blood[37], but upfront depletion of rRNA is not desirable when using very low starting amounts of RNA[38] and is prone to increase the failure rate of ligation-based library preparation methods[14]. Of note, both Rasmussen[36] and Moufarrej[15] used the SMARTer Stranded Total RNAseq Pico Input Mammalian kit (Takara), but Rasmussen omitted the rRNA depletion step and employed a target capture method. To mitigate these challenges, we applied several approaches. First, in our case-control design we paired each case with up to 4 controls by matching on maternal body mass index (BMI), fetal sex, maternal smoking, maternal age, and the sampling time to avoid any confounding effects. We ensured that no controls were shared between cases. Second, in the cfRNA capture and library preparation steps, we supplemented an off-the-shelf enrichment kit (Illumina Nextera Flex, now known as the Illumina RNA Prep with Enrichment) with an additional bespoke enrichment method using probes specific to the human placenta transcriptome[31]. cfRNAs were quantified against our human placenta transcriptome to reflect the fact that the disease is considered a placental disorder.

Third, our RNA-seq dataset is far more deeply sequenced than the existing studies, with an average of ~300 million reads per sample. In compassion, the datasets from Rasmussen and Moufarrej generated ~20 million reads per sample, and the dataset from Munchel, ~40 million reads per sample. Also, we employed a number of quality control metrics such as the percentage of mapped reads, the extent of spliced alignment, the percentage of rRNA mapped reads, the percentage of reads mapped to intronic and intergenic and so on (Supplementary Data 13). All the RNA-seq samples we analysed had >90 million mapped reads and >18% of spliced alignment. Finally, for data processing, we carefully selected our bioinformatics pipeline based on the performance of predicting fetal sex by measuring the extent of chromosome Y encoded transcripts, and we chose the pipeline which favoured a smaller number of false positives and false negatives (see Supplementary Information - Selection of RNA-seq quantification method). Finally, we used multiple approaches to identify differentially expressed genes and compared 11 ML methods to ensure that the final selection of cfRNA biomarkers was well justified. This combination of features in the experimental design is in contrast to other studies as is our finding that quantitation of only 2 cfRNAs is sufficient to predict PE with FGR. This may indicate that with the appropriate design, a simple model is sufficient to make meaningful prediction.

In conclusion, our findings open a new window of liquid biopsy by using cfRNAs for monitoring complications of human pregnancy, and we have demonstrated this for the combination of PE with FGR.

## Methods

### Study design and sample collection

All the samples were obtained from the POP study, a prospective cohort study of nulliparous women attending the Rosie Hospital, Cambridge (UK) for their dating ultrasound scan between 14 January 2008 and 31 July 2012. The study has been previously described in detail[18,19]. Ethical approval for the study was given by the Cambridgeshire 2 Research Ethics Committee (reference number 07/H0308/163) of the NHS Health Research Authority and all participants provided written informed consent. Preterm was defined as <37 weeks of gestational age (wkGA) and term was defined as ≥37 wkGA. Cases of PE were defined on the basis of the 2013 ACOG criteria[39] and cases of FGR had a customized birth weight <10th percentile[40] for preterm FGR and a population-based birth weight <10th percentile[41] for term FGR. Controls were defined as pregnancies resulting in a live-born infant with a birth weight percentile in the normal range (20–80th percentile[42]) with no evidence of slowing in fetal growth trajectories, and with no evidence of hypertension at booking and during pregnancy, PE, hemolysis/elevated liver enzymes/low platelet (HELLP) syndrome, gestational diabetes or diabetes mellitus type I or type II or other obstetric complications. From among eligible non-cases, we excluded women who had missing data on body mass index (BMI) at 12wkGA, smoking at 12wkGA or fetal sex, women who lacked one or more samples at 12-36wkGA or sampling was performed outside the acceptable GA range. Acceptable sampling GA ranges were: 12 week sampling: 10– <15 wkGA, 20 week sampling: 18– < 23 wkGA, 28 week sampling: 26– <31 wkGA and 36 week sampling: 34- <39 wkGA.

Each case was paired up to 4 controls uniquely, i.e., no controls will be shared between cases, by matching the followings in the order of importance: 1) fetal sex (exact), 2) smoking (exact), 3) maternal BMI

(as close as possible), 4) maternal age (as close as possible), and 5) exact GA at 12 week sampling (as close as possible, ideally within a week). Matching was performed using Stata version 15 for each case from round 1 (the most stringent matching criteria) up to round 29 (the most relaxed matching criteria) as necessary until all four matched controls were identified (see Supplementary Data 1 for descriptive statistics of matched cases and controls). A total of 195 patients were considered in this study, among which there were 39 cases of PE with FGR, and 156 control subjects. Among the 39 cases, 15 were from the preterm and 25 were from the term delivery. The 15 preterm cases were matched to a total of 60 controls and the 24 term cases were matched to a total of 96 controls.

## Sample collection

All blood samples were collected in EDTA-tubes (S-Monovette 7.5 ml, K3E from Sarstedt, cat 01.1605.001) and spun at 1610 x g at 4 °C for 10 minutes. If delay in spinning was inevitable plasma samples were kept on ice. Samples were centrifuged within 30 minutes of collection. After centrifugation, each plasma sample was divided in 4 aliquots and stored in −80 °C. All samples used in this study have not previously been thawed. All lab work was carried out by operators blind to case/control status. Sample processing was performed in batches of 20 during RNA extraction and batches of 40 during library preparation such that all 4 plasma samples from each case and all 4 matching controls would be run in the same batch. Samples from a case at a given gestational age were analysed in the same batch as the 4 matched controls also obtained at the same gestational age.

## cfRNA extraction and library preparation

Samples were processed into sequenceable libraries by multiple operators in the Illumina laboratory Services (ILS), Foster City, CA. Each batch was randomized by the investigators, and disease status was blinded to the operators who processed the samples. cfRNA was extracted from 2 ml of plasma per donor using the Circulating Nucleic Acid Kit (Qiagen, P.N. 55114) following the manufacturer's protocol. The only deviation was omitting the addition of carrier RNA to the ACL buffer. The isolated RNA underwent DNase I treatment (ThermoFisher Scientific, P.N. AM2239) to eliminate contaminating DNA, followed by purification using RNAClean XP beads (Beckman Coulter, P.N. A63987). The samples were converted into sequenceable libraries using the Illumina RNA Prep with Enrichment Kit (Illumina, P.N. 20040537). As outlined in the protocol, cfRNA is converted into double-stranded cDNA using random hexamer primer synthesis. The cDNA is tagmented and adapter sequences and unique dual indexes are incorporated during index PCR. Finally, libraries were quantified with the AccuClear Ultra High Sensitivity dsDNA Quantitation Assay Kit and pooled (four libraries at 200 ng each) for enrichment.

## Whole-transcriptome enrichment

Exonic enrichment was performed using the Illumina RNA Prep with Enrichment Kit, targeting transcripts originating from exons within sequencing libraries. Biotinylated probes from the Illumina Exome Panel (Illumina, P.N. 15034576) hybridized with the libraries, and subsequent capture with streptavidin beads enriched the exonic content. PCR amplification ensured sufficient yield for sequencing, followed by library quantification and pooling. Two modifications were made to the protocol: first, 1996 additional probes targeting placental exons (including circular RNAs) not covered by the standard panel were included. These additional probes were identified from RNA-seq data of 295 placental samples and generated using Illumina's DesignStudio. Second, to minimize the presence of highly abundant haemoglobin genes (*HBA1*, *HBA2*, *HBB*), reverse-complementary blockers were incorporated into the enrichment reaction.

## Sequencing and quality control

For in-depth analysis, each library underwent 50 base paired-end sequencing on an Illumina NovaSeq 6000 platform with an S4 flow cell. Batches of 96 libraries were pooled together for sequencing. We generated an average of ~300 million reads per sample. For an initial quality control (QC) of sequencing reads, we ran Illumina's DRAGEN™ (Dynamic Read Analysis for GENomics) RNA pipeline (v3.8) to perform alignment, transcript quantification, and general QC metrics such as the number of duplicated reads, the per-centage of reads assigned to the ribosomal RNA (rRNA), introns and exons. We also ran preseq[43] to estimate the complexity of the sequencing library by checking the total size of the population (via bound_pop module) and RSeQC[44] to identify gDNA contamination (via RSeQC bam_stat.py script). We set up three levels of QC metrics: FAIL, WARN, and PASS. Samples were flagged as FAIL if the number of mapped reads were <90 million or the per-centage (%) of spliced alignment were <18.9. Samples were flagged as WARN if the number of mapped reads were <100 million or manually checking the distribution.of the following measures: 1) % of properly mapped reads, 2) % of rRNA mapped reads, 3) the library complexity, 4) the number of genes detected with ≥1 count per million (CPM), 5) % of reads mapped to intron region, and 6) % of reads mapped to intergenic region. Samples were flagged as PASS otherwise. Three samples were flagged as FAIL due to low % of spliced alignment – we reprocessed the assays and rescued one (GS-179-HQ) but discarded two (GS-59-CX and GB-B-374-UW) as they retained the same FAIL flag and were dropped from the subsequent analysis. 16 samples were flagged as WARN but they were retained in the analysis as there was no sign of systemic bias (e.g., no clear indication of batch effect). Supplementary Data 13 shows the sequencing statistic and the QC metrics in details.

## RNA-seq data processing

We selected an RNA-seq quantification method based on a benchmark of several approaches. We chose the most reliable approach based on the performance of predicting fetal sex by measuring the extent of chromosome Y encoded transcripts (see Supplementary Information and Supplementary Fig. 1 for detail). Based on this benchmark result, we chose Salmon (v1.5.2)[45] in mapping-mode to process our RNA-seq datasets. For the reference of transcript annotation, we re-constructed placenta transcriptome based on 295 placenta RNA-seq datasets[31] using StringTie (v2.2.1)[46] and Ensembl (v88)[47]. Transcripts having at least 10 bp depth of coverage (-c 10) and at least 5 bp junction reads (-j 5) were considered. We used both known and novel transcripts supported with a minimal level of expression (i.e., ≥0.1 TPM in ≥10% of cohort). A total of 82,980 transcripts were used for a decoy-aware transcriptome indexing by Salmon. Finally, the abundances of transcripts were quantified by the following pseudo-command of Salmon: salmon quant -p 32 -i POPS_TR_INDEX -l A -1 S1_FQ_1.fq -2 S1_FQ_2.fq --seqBias --gcBias --posBias --discardOrphanQuasi --writeUnmappedNames --writeMapping -o OUTPUT_DIR | samtools view -bS > OUTPUT_DIR/S1_salmon.bam.

## Differentially expressed gene analysis

The differentially expressed gene analysis in a case-control setting was conducted for each gestational epoch (12wk, 20wk, 28wk, and 36wk) separately, except for the 36wkGA gestation samples of the pre-term dataset which has only one PE case. The transcript-level read count matrices (e.g., quant.sf files) were imported using tximeta (v1.8.5) Bioconductor package[48] and merged at the gene-level. We only considered genes found in ≥10% of samples (i.e., ≥23) having ≥10 reads, and discarded genes detected as dispersion outliers by DESeq2 (v1.30.0)[20]. A total of 15,150 genes and 221 samples were used to find differentially expressed genes by adjusting the batch number, the fetal sex, and the gestation of the samples in the design matrix of DESeq2. The *p*-values were calculated from the null hypothesis that the fold

changes were less than or equal to 20% (i.e., lfcThreshold=log2(1.2)) in case and control groups. For edgeR (v3.32.1) analysis[21], we used makeDGEList function of tximeta Bioconductor package to convert the data object of the 15,150 genes across the 221 samples as mentioned above. The gene-level count matrix was normalised by using calcNormFactors function of edgeR with TMM method (trimmed mean of M values) and a quasi-likelihood negative binomial generalised log-linear model (i.e. glmQLFit) was applied to account for the batch number, the fetal sex and the gestation information in the design matrix of edgeR. For a statistical test, we used glmTreat of edgeR[49] with at least 20% fold-change (i.e., lfc=log2(1.2)). For the validation dataset, we repeated the same aforementioned procedure by restricting the 15,150 genes considered in the discovery dataset. For the gestational age-related differentially expressed gene analysis, we used a total of 96 healthy control samples from the discovery dataset (i.e., validation cohort) and applied the same threshold (i.e., the fold change less than or equal to 20%) for the test at each gestational interval (12wk to 20wk, 20wk to 28wk, and 28wk to 36wk) using DESeq2 only. For both DESeq2 and edgeR, to control false discovery the p-values were corrected by the Benjamini and Hochberg method[22] and they were conducted using Bioconductor v3.12 and R v4.0.3[50].

### Data transformation and univariable logistic regression

The gene-level count matrix was converted as the unit of CPM, in log2-scale, via the cpm function of edgeR and it was further transformed into a matrix of the z-score using the mean and standard deviation of logCPM from the control samples of each corresponding gestational age group (i.e., 12, 20, 28 and 36 weeks). Using the binary outcomes, (i.e., case and control status) as dependant variables and the z-scores as independent variables, we applied a generalised linear model for each of the 15,150 genes using the glm function of R stats package (v4.0.3). Both the Akaike information criterion (AIC) and the Bayesian information criterion (BIC) were obtained from the corresponding univariable model and the area under the ROC curve (AUC) was calculated using the pROC package (v1.17.0.1)[51]. The p-values were calculated against the null hypothesis that is the odds ratio is equal to 1 and they were adjusted for multiple comparisons using Benjamini and Hochberg method. The distribution of the p-values was tested against a uniform distribution using one-sample Kolmogorov-Smirnov test.

### Selection of shared differentially expressed genes

To select a subset of genes that best explains the outcome of samples from the 28wkGA group of the discovery cohort, we used the following four criteria: 1) the p-values from DESeq2, 2) the p-values from edgeR, 3) AIC, and 4) AUC from univariable logistic regressions. Then we selected the top 1% genes for each category (i.e., 151 genes of the lowest p-values from DESeq2 and edgeR, 151 genes having the lowest AIC, and 151 genes having the highest AUC) and constructed a Venn diagram using ggvenn (v0.1.0) R package[52]. We selected a total of 17 genes satisfying all the four criteria (i.e., the intersection) and constructed a gene expression matrix (i.e., the 17 genes across the samples from the 28wkGA group of the discovery cohort) using the z-score.

### 5-fold cross validation with 5 repetitions

We considered a total of 11 ML methods (Table 2) to select the best performing method based on the 5-fold cross-validation (CV) with 5 repetitions. We randomly split the samples into 5 strata by distributing the number of case and control outcomes as even as possible across the 5 folds. This stratified 5-fold splitting was repeated 5 times by changing a seed number in each repetition, and the 11 ML models were trained to choose a desired number of predictors from 2 to 6. In glParallel[27], a brute-force exhaustive search method, for a given number of predictors, it searched all possible combinations of predictors in multivariable regression models and picked the best model based on the highest predictive performance. For example, glParallel trained a

total of 2380 models, which is the possible number of combinations having 4 predictors out of 17, and chose the best model based on the highest Leave Pair Out Cross Validated (LPOCV) Area Under the ROC Curve (AUC), a version of optimism-corrected AUC[28]. In LPOCV-AUC, a model was fitted based on a given set of training samples except one pair of case-and-control, then the model was used to predict the outcome of the remaining pair. The LPOCV-AUC was calculated as the proportion of all pairwise combinations in which the predicted probability was greater for the case than for the control. Penalised regression methods (ENet and LASSO) were initially fitted using the train function for Elastic net method and the cv.glmnet function for LASSO, from the caret (v6.0.94) and the glmnet (v.4.1.2) R packages, respectively. For ENet1, both the parameter $\alpha$ and $\lambda$ were tuned by the caret::train, whereas the parameter $\lambda$ was further tuned by the glmnet::cv.glmnet for ENet2, which is the one presented in Fig. 3A. Next, based on the best fitted penalised regression models, a matrix of the $\beta$ coefficients was examined to find the first set of predictors with non-zero $\beta$ coefficients that satisfied a desired number of predictors. If the number of predictors with non-zero coefficients exceeded the desired number, the absolute values of coefficients were sorted in their decreasing order and only the desired number of predictors were selected with their highest absolute scores. For the remaining methods, except mSVM-RFE[53] which embedded a Recursive Feature Elimination (RFE) algorithm internally, we used the caret::rfe function by controlling the sizes parameter to have the corresponding models with the desired number of predictors. Having selected a set of predictors for each of the 11 ML methods, a logistic regression model was fitted on the training fold based on the selected predictors, and its predictive performance, i.e. the AUC, was calculated using the remaining held-out test fold. As the 5-fold CV was repeated 5 times, the 25 cross-validated AUCs were averaged by taking the mean AUC. This procedure was repeated from a selection of 2- to 6-predictor models, i.e., 5 times, so the cross-validated AUCs were again averaged by taking the mean values of AUCs. Having identified the best method (as shown in Fig. 3A), it was applied to choose 2 to 10 predictors using all the 28wkGA samples from the discovery dataset and the selected predictors were used to fit multivariable logistic regression models using the same training dataset (Fig. 3B). Finally, we evaluated the predictive performance of those 2- to 10-predictor logistic regression models using the term validation cohort and the external Munchel dataset (Fig. 4).

### Processing external validation dataset

For an external validation, we downloaded the raw sequencing counts file (Data File S2: Raw whole-transcriptome sequencing counts for iPEC cohort; n = 113) from Munchel et al.[14]. The Data File (in the excel file format) was read and parsed by using readxl (v1.3.1) and data.table (v1.13.6) R packages, respectively. For downstream processing, we only considered those genes in the final set of 15,150 genes that were used in the differentially expressed gene analysis of the discovery and the validation cohort (see above). Using the filtered gene-level raw count matrix, we ran edgeR and constructed a matrix of CPM, in log2-scale, via the cpm function of edgeR. The matrix was further transformed into a matrix of the z-score using the mean and standard deviation of logCPM from the 73 control samples.

### Longitudinal analysis of cfRNAs

We modelled the trend of cfRNA abundance by gestational age using lme4 (v1.1.37) R package. We set up a mixed effect model for each cfRNA by taking a random intercept in the model which allows different baseline levels for each participant. For fixed effects, we considered both linear and quadratic term of gestational age as fixed effects. We scaled the gestational age (in week) by subtracting 4 weeks from the actual gestational age, then divided by 8 weeks. This transformed the gestational time of 12wk, 20wk, 28wk and 36wk to 1, 2, 3,

and 4, respectively. We obtained the *p*-values of fixed effects from lmerTest (v3.1.3) R package and the *p*-values from the linear and quadratic term were combined by the Fisher's method[54] using a chi-squared distribution, followed by correcting for multiple tests using Benjamini and Hochberg method[22]. The marginal and conditional coefficient of determination were calculated using MuMIn (v1.48.4) R package.

## Reporting summary

Further information on research design is available in the Nature Portfolio Reporting Summary linked to this article.

## Data availability

The maternal plasma RNA-seq data have been deposited in the European Genome-phenome Archive (EGA) under accession number EGAD00001015416. Data access is restricted to comply with the ethical approval relating to these samples, which requires that data to be shared are anonymized. The EGA requires those requesting data to complete a Data Access Agreement (DAA) which stipulates that users will not attempt to de-identify the participants. Access to the POPS genomic data can be requested via the Data Access Committee number EGAC00001000582. The EGA provide the following Data Access Agreement for the POPS data (https://ega-archive.org/assets/files/Data%20Access%20Agreement%20EGA%20and%20Application%20Form%20-%20TEMPLATE.docx). Requesters should expect a reply within 2 weeks of submitting a request and there is no time limit on availability once data access has been authorized.

## Code availability

All original code is publicly available at https://obsgynaecam.github.io/cell-free-rna-2024/[29].

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

## Acknowledgements

We would like to thank Katrina Holmes and Josephine Gill, for technical assistance during the study and also Cephas Andoh, Ahbby Garcia, Silvia Chan, Kristina Kwock and Ploy Setthasap, all from Illumina laboratory services for their assistance. Figure 1 was created in BioRender. Gong, S. (2025) https://BioRender.com/bwfj19q. The views expressed are those of the authors and not necessarily those of the NHS, the NIHR, or the Department of Health and Social Care. This work was supported by the Medical Research Council, United Kingdom G1100221 and MR/K021133/1 (G.S.); the National Institute for Health Research (NIHR) Cambridge Biomedical Research Centre (Women's Health theme) (G.S.) and Illumina (F.K.).

## Author contributions

Conceptualization: S.G., S.S., F.K., G.S., D.C.-J.; Methodology: S.G., C.R.-H., S.S., G.S., D.C.-J.; Software: S.G.; Validation: S.G., S.R., C.R.-H.; Formal analysis: S.G.; Investigation: C.R.-H., S.R., J.Y.W., K.S., E.C.; Resources: G.S., D.C.-J., S.R., J.Y.W., K.S.; Data Curation: S.G., S.R., U.S., E.C.; Writing - Original Draft: S.G., G.S., D.C.-J.; Writing - Review & Editing: C.R.-H., S.R., U.S., F.K., G.S., D.C.-J.; Visualization: SG. Supervision: C.R.-H., J.Y.W., K.S., F.K., G.S., D.C.-J.; Project administration: S.S., GS, D.C.-J.; Funding acquisition: F.K., G.S., S.C.-J.

## Competing interests

G.S. and D.C.-J. have received research support from Roche Diagnostics Ltd and Sera Prognostics (fetal growth restriction, preeclampsia and preterm birth). G.S.'s department has received payment from Roche for a talk given by G.S. (fetal growth restriction). GS has been a paid consultant to GSK (preterm birth) and is a member of a Data Monitoring Committee for GSK trials of RSV vaccination in pregnancy. G.S. is currently a member of a Data Monitoring Committee for RSV vaccination in pregnancy (Moderna) and chairs a Data Monitoring Committee for a hyperemesis gravidarum therapeutic trial (NG Biopharmaceuticals). G.S. and D.C.-J. have received support from Pfizer (outside the scope of this work). C.R.-H., S.R., J.Y.W., K.S., S.S. and F.K. are, or were at the time of this work, employees of Illumina. The remaining authors report no conflict of interest.
