## [Peer Review file · Nature Communications]

Raised Leptin and Pappalysin2 cell-free RNAs are the hallmarks of pregnancies complicated by preeclampsia with fetal growth restriction

Corresponding Author: Professor D. Stephen Charnock-Jones

Version 0:

Reviewer comments:

Reviewer #1

(Remarks to the Author)

RESEARCH SUMMARY:

In their manuscript Gong et al report on the use of cfRNA as present in maternal plasma to predict pregnancies complicated by the combination of Preeclampsia and Fetal Growth Restriction ([PE+FGR]). Using biobanked maternal plasma samples prospectively collected in the POP study, the authors conducted a 1:4 case:control study comprising longitudinally collected samples (up to 4 sampling timepoints across gestation) of 39 women developing [PE+FGR] in their pregnancy and 156 control pregnancies, i.e., women who didn't experience obstetric complications during pregnancy. The study population was split into a discovery and (internal) validation cohort, whereby the cases (n=15) in the discovery cohort consisted of women requiring delivery for PE+FGR before 37 weeks of gestation (preterm delivery), and the cases (n=24) in the (internal) validation cohort consisted of women requiring delivery for PE+FGR from 37 weeks onward (term delivery).

Based on the levels of cfRNA in patient samples collected at ~ 28 weeks of gestation, the authors selected 17 genes of interest based on their ability to differentiate cases from controls, i.e, the predictors. Following an exploratory evaluation of 11 machine learning methods based [PE + FGR] prediction model performance, elastic net was used to develop final prediction models with n=2 to n=10 predictors in the 28 weeks discovery data.

[PE+FGR] prediction performance (AUC) of the final models was observed in the earlier gestational age windows (12, 20 wks) in the discovery cohort and in the 4 gestational age data sets (12,20,28,36 wks) of the internal validation cohort.

Prediction performance increased across the gestational age data sets, from virtually non-predictive at 12 weeks to moderately predictive or diagnostic at 28 weeks and 36 weeks. It was found that the 2 predictor model based on the cfRNA levels for PAPP2 and LEP were most performant as compared to the models with more predictors (n>2). Interestingly, the individual predictors, PAPP2 and LEP can have equivalent stand-alone prediction compared to the model; this is true for PAPP2 at all gestational age windows and LEP at 36 weeks.

Finally, the authors applied the [PAPP2 + LEP] prediction model to the cfRNA data from Munchel et al (1). This study involved samples taken at ca. 30.5 wks of gestation from 40 early onset preeclampsia cases (preeclampsia diagnosis before 34 week of gestation) with (45%) or without (55%) FGR and 73 control pregnancies. Remarkably, PAPP2, LEP and the [PAPP2 + LEP] prediction model were highly diagnostic in the Munchel data set with AUCs of ca. 0.95.

RESEARCH REVIEW

A) VALIDITY OF THE RESEARCH: In general terms the manuscript & supplementary website <https://obsgynaecam.github.io/cell-free-rna-2024/> of Gong et al covers in adequate detail the research conducted. The authors should consider improving transparency about the fact that essentially only the cfRNA data from the 28 weeks discovery samples were considered for model development and no longitudinal modelling was performed.

B) SIGNIFICANCE OF THE RESEARCH: The manuscript lack clarity in its objective. At least three objectives are identifiable to the reviewer

a. Demonstration that the authors cfRNA analytical and data analysis work flow is more robust/more fit-for-purpose than any of the recent cfRNA publications in the same space >> as presented there is no convincing argumentation provided that the key technical changes, i.e, study design, use of latest NGS (illumina) tools, sequencing depth, the selection of a particular cfRNA quantification method, the application of the discrete super learner approach to select a machine learning method, are more appropriate than workflows reported by others for the same / similar clinical applications (1-4). For instance, it is

unclear why the authors did not consider AdaBoost as used by Munchel et al in their super learner approach.

b. Demonstration that circulation PAPP2- and/or LEP- cfRNA levels have clinical application potential in monitoring the development or pregnancy complications >> Justification of the selection of PE+FGR as the clinical application. The introduction appears to set up the narrative that the clinical application is about identifying cfRNA biomarkers for placental insufficiency, i.e., the intersect of FGR and (early onset) PE. Monitoring pregnancy for placental insufficiency would have clinical relevance. However, clarity of the intended clinical application is not maintained throughout. If the authors would to highlight clinical application potential, benchmarking PAPP2 and LEP cfRNA prediction against circulating s-Flt1 and PIGF will be instructive; the POP study likely has this data readily available (5,6).

c. Independent validation of PAPP2- and LEP- cfRNA results as biomarkers for preeclampsia as reported by others >> In view of the variability in analytical work ups and data analysis pipelines, the work of Gong et al. builds the body of scientific evidence for these markers; this is certainly a significant observation.

Identification and clear messaging of what the authors consider the main scientific contribution of their work adds will improve impact of their manuscript.

C) DATA AND METHODOLOGY: In general Gong et al provided high level detail in terms of study design, sample work up, cfRNA analysis, data processing, statistical analysis and model development. The quality of the dedicated website materials (<https://obsgynaecam.github.io/cell-free-rna-2024/>) is remarkable. Whereas the code was not tested / verified, review of the within code annotation suggests that the code was developed by (a) skilled scientist(s). Some sporadic miss spellings were observed: f.i., gestion instead of gestation, shink instead of shrink,...

The manuscript can be improved by addressing the following items:

a. Salmon is selected as the preferred RNA-seq quantification method based on a thorough evaluation of correctly calling fetal sex based on chromosomal Y transcripts. Yet, it remains unexplained why the correct calling of fetal sex is indicative for Salmon being the better method for comprehensively identifying / quantifying cfRNA over the other methods.

b. Selection of the so-called core genes is based on an overlap of several ranking exercises. Without rationale which eg refers to (different) underlying statistical assumptions used in DESeq2, edgeR and the AUC and AIC statistics as derived from the same univariable logistic regressions, this selection appears largely arbitrary and doesn't justify labelling the 17 genes as "core genes" as this may infer they are "core" to development of placental insufficiency; without justification of the latter, less "leading" terminology should be adopted.

c. When it comes to the use of the discrete super learner approach: The ranking criteria (Rank Sum and Mean Rank) remain unexplained, albeit they can be inferred from the results in supplementary data #6. Based on the statistics presented in Fig 3, panel A, it is not apparent to the reader whether there is a clear-cut best ML, especially when one accounts for the fact that the final (robust) model put forward is a simple 2 predictor model. Naming the genes selected in the different models in supplementary data #6 in the same fashion as supplementary data #7 would be helpful to the reader to assess to what extent the different ML select the same predictors. Based on the data presented, the application of the discrete super learner approach appears of limited added value in this study; the authors may want to argue otherwise.

As mentioned earlier, explanation of the non-consideration of adaboost as used by Munchel et al would also be helpful (e.g. only using non-ensemble ML techniques?).

d. Given the simplicity of the final model, it would be helpful to provide plots (in supplementary) of the z-scores for PAPP2 and LEP with case and control labels in a simple x-y scatter for each of the predictions in Figure 4 – Panel D.

D) ANALYTICAL APPROACH:

a. The practical experimental design as outlined in the Sample collection paragraph is an ideal preparation for establishing longitudinal intra-patient data as for each patient all gestational timepoints are processed in the same batch. This begs the question why no longitudinal data analysis / modelling was planned/performed by the authors; please comment.

b. Development of robust multivariable predictors is constrained by the number of cases in the discovery cohort. Given the low number of effective cases (n=13 at 28 weeks, see tabulation of discovery cohort on <https://obsgynaecam.github.io/cell-free-rna-2024/sample.html>), the choice for a final 2 predictor model is appropriate. The authors may want to consider re-analysing the data without splitting the data set and use the complete POP (28 weeks) sample set for model development as advised (7). This may lead to a robust n>2 predictor model with better performance and side step the risk that the discovery cohort and the internal validation cohort represent possibly different disease subtypes, as early onset presentations of placental insufficiency (<34 weeks of gestation) is oftentimes considered to be distinct to later presentations of placental insufficiency. The applied study population split may therefore be biased to select for early onset disease markers. It is noted that the Munchel external validation cohort is an early onset PE +/-FGR cohort, which may explain why the PAPP2 + LEP model has better performance in Munchel as compared to the internal validation cohort. It is observed that the authors did not explicitly mention this feature of the Munchel study population.

c. It is noted that the 2 predictors selected for the model are the 2 only predictors (2/17) which are upregulated in the PE+FGR cases (table 1). Is this to be expected or is there a possible predictor selection bias in the ML chosen. Please comment.

E) CLARITY AND CONTEXT: The transparency of the manuscript would benefit from clarifying the following

a. Selection of PE+FGR as the clinical application. The introduction appears to set up the narrative that the clinical application is about identifying cfRNA biomarkers for placental insufficiency, i.e., the intersect of FGR and (early onset) PE. This clarity is not maintained throughout.

b. There is conflation of PE and FGR in terms of maternal presentation in lines 50 – 54 ; endothelial dysfunction is considered a hallmark of PE and not FGR.

c. The discovery cohort is effectively a 28 weeks only cohort: DEGs were only selected from this timepoint; model development only considered this timepoint. The longitudinal nature of the data was not leveraged for model development. Performance of the 28 weeks model is solely evaluated at the different timepoints. The lack of performance in the earlier gestational ages may indicate that the composition of the cfRNA repertoire at 28 weeks is different than these at earlier

timepoints, making the model ill-suited for risk prediction in the first half of pregnancy. Longitudinal analysis of cfRNA / gene expression may have elicited true monitoring

- d. Add a brief clarification that only/mainly the mRNA fraction of cfRNA is being explored, this would improve readability as the manuscript switches regularly between cfRNA and mRNA terminology
- e. The Benjamini and Hochberg correction is better characterised as a method to control false discovery rate than a multiple-test correction
- f. The authors benchmark their data to literature investigating use of cfRNA for prediction/diagnosis of preeclampsia. No attempt was made to benchmark to FGR literature; cf. ref 14 in Gong et al.
- g. In absence of data on PE without FGR and FGR without PE, the question remains what pregnancy complication the reported cfRNA signatures are specifically reflecting. The authors are invited to briefly comment on this in the Discussion. See also earlier comments on clarity of clinical application.

REFERENCES

1. Munchel S, Rohrbach S, Randise-Hinchliff C, Kinnings S, Deshmukh S, Alla N, et al. Circulating transcripts in maternal blood reflect a molecular signature of early-onset preeclampsia. *Sci Transl Med.* 2020;12(550).
2. Rasmussen M, Reddy M, Nolan R, Camunas-Soler J, Khodursky A, Scheller NM, et al. RNA profiles reveal signatures of future health and disease in pregnancy. *Nature.* 2022 Jan 20;601(7893):422–7.
3. Moufarrej MN, Vorperian SK, Wong RJ, Campos AA, Quaintance CC, Sit R V., et al. Early prediction of preeclampsia in pregnancy with cell-free RNA. *Nature.* 2022;602(7898):689–94.
4. Ngo TTM, Moufarrej MN, Rasmussen MLH, Camunas-Soler J, Pan W, Okamoto J, et al. Noninvasive blood tests for fetal development predict gestational age and preterm delivery. *Science (80-).* 2018;360(6393):1133–6.
5. Gaccioli F, Sovio U, Cook E, Hund M, Charnock-jones DS, Smith GCS. Screening for fetal growth restriction using ultrasound and the sFLT1 / PIGF ratio in nulliparous women : a prospective cohort study. *Lancet child Adolesc Heal.* 2018;2(8):569–81.
6. Sovio U, Gaccioli F, Cook E, Hund M, Stephen Charnock-Jones D, Smith GCS. Prediction of Preeclampsia Using the Soluble fms-Like Tyrosine Kinase 1 to Placental Growth Factor Ratio: A Prospective Cohort Study of Unselected Nulliparous Women. *Hypertension.* 2017;69(4):731–8.
7. Collins GS, Dhiman P, Ma J, Schlüssel MM, Archer L, Calster B Van, et al. Evaluation of clinical prediction models (part 1): from development to external validation. 2024;(part 1).

ADDENDUM - Overview of the research conducted

The cfRNA analysis involved

- nucleic acid extraction;
- RNA purification and conversion into sequence-able libraries;
- exonic enrichment using Illumina Exome panel augmented with probes targeted to placental exons and use of reverse-complementary blockers for highly haemoglobin genes to limit their representation in the libraries;
- pooling of 96 Libraries for 50 base-pair ended NGS using Illumina NovaSeq 6000 with S4 flow cell.
- extensive NGS read Quality Control which led to excluding 3 patient samples.
- Used of fetal sex prediction based on Y chromosome transcripts to select a quantification approach for RNA-seq quantification

[PE+FGR] Prediction model development involved

- using the RNA transcript read levels to select the genes for analysis (n = 15,150 based on less than 90% missing rate in discovery patient samples across 3 gestational age sampling windows (12,20,28 weeks) and at least 10 reads / gene);
- identification of differentially expressed genes between the cases and controls of the discovery cohort at three discrete gestational age sampling windows (12,20,28) using 2 methods (DESeq2 and edgeR) and application of Benjamini-Hochberg control of False Discovery Rate
- logarithmic conversion of the gene quant matrix and transformation in z-scores per gestational sampling window.
- univariable discrimination analysis (AUC) between the cases and controls of the discovery cohort at three discrete gestational age sampling windows (12,20,28) using matrix of z-scores, with Benjamini-Hochberg control of False Discovery Rate.
- 2 stage selection of differential expressed genes of interest for the 28 weeks gestational age timepoint leading to a set of 17 genes (“predictors”) for prediction model building using the 28 weeks gestational age timepoint data of the discovery cohort
- Performing an exploratory prediction model development analysis using 11 machine learning (ML) methods with the aim of selecting the most performing ML method. This ML assessment involved the 28 weeks gestational age timepoint “gene level” data of the discovery cohort, generation of logistic regression models with n= 2 to n=6 predictors for any of the 11ML under evaluation using a 5 x 5-cross validation approach
- Using the chosen ML method (elastic net), prediction models with n=2 to n=10 predictors were fitted to the 28 weeks gestational age timepoint “gene level” data of the discovery cohort. The 2 predictor model [PAPPA2 + LEP] was put forward as the most robust / least overfitted.

[PE+FGR] Prediction model evaluation / validation performed

- Prediction performance of the 28 weeks [PAPPA2 + LEP] model as well as the individual predictors PAPPA2 and LEP were observed in the other discovery cohort gestation time windows (12, 20 wks) as well in the 4 gestational age time windows available for the internal validation cohort (12,20,28, 36 wks).
- External validation of the prediction performance of the 28 weeks [PAPPA2 + LEP] model as well as the individual predictors PAPPA2 and LEP was obtained by re-analysis of the cfRNA data set from Munchel et al (1) consisting of samples obtained from women diagnosed with early onset preeclampsia (preeclampsia before 34 weeks of gestation) and control pregnancies.

Additional analyses performed involved

- Evaluation of the PAPPA2 and LEP mRNA levels for the case and control groups across the various gestational windows

in the POP discovery and validation cohorts as well as the Munchel data set.

Overview of the discussion

- Recapitulation of the exceptional performance of the 28 weeks [PAPPA2 + LEP] model in the Munchel data set, positing the markers as diagnostic.
- Biological plausibility of the placenta as the source of the observed association of increased levels of cfRNA PAPPA and LEP levels as PE+FGR outcome.
- Benchmarking marker findings to other cfRNA studies for prediction of preeclampsia. Significant overlap in the selection of differentially expressed genes as derived from the 28 weeks samples from the discovery cohort was noted, mostly with the Munchel data which (also) involved late pregnancy samples. Overlap with cfRNA studies using samples obtained early in pregnancy is less.
- Comparison of experimental and data analysis pipeline with earlier studies to rationalise different cfRNA findings across studies.

(Remarks on code availability)

The code as available at the provided supplementary website (<https://obs gynae cam.github.io/cell-free-rna-2024/>) was visually inspected; the within code annotation was comprehensive.

No attempts were made to install and run code.

The quality of information provided at supplementary website was high.

Reviewer #2

(Remarks to the Author)

Here, Gong et al present a new study of using cfRNA to predict preeclampsia and fetal growth restriction. Liquid biopsies hold promise as a minimally invasive approach for monitoring processes inside the body. To date, most studies have focused on cfDNA so it is my opinion that cfRNA holds more opportunities. Thus, this study presents an interesting use of cfRNA monitoring and strong results regarding the ability to predict outcomes. Given the large number of ML models explored, it is clear that the signal is very robust. The generalization to the Munchel et al study is also reassuring. Moreover, the authors deserve credit for the Github repository containing the code to reproduce the results. I wish that more authors put in a similar effort for documenting their work. Nevertheless, I believe that the following major concerns must be addressed:

1. I do not understand the point of fig 2a? Why is it meaningful to test the distribution of uncorrected p-values? In my opinion, there is little value in examining the distribution of p-values. As far as I am concerned, there is no interesting or meaningful model stipulating how they should be distributed, so I do not see the point of this analysis or test.
2. I am curious about the gIParallel model with two genes in fig 3A. I understand that overall this model performs poorly. However, given that the authors later focus on models with two genes, I am wondering if this model could be useful after all. I encourage the authors to investigate, in case it turns out that they have missed a high-performing model.
3. How about using partial AUC (pAUC) instead of AUC for ranking the models? In my opinion, AUC may put too much emphasis on regions of the ROC curve that are not relevant in practice. For example, if the intention is to run the pregnancy test with high specificity (I do not know if this is the case since I am not a clinician), then using a pAUC with a cut-off at for example 0.2 on the x-axis might be more appropriate. This could potentially result in some of the method rankings changing.
4. If I understand correctly, for most participants in the study there is cfRNA from more than one time point. However, this information does not appear to be used at all in the analysis. I would be interested for example for fig 5 to learn if the LEP and PAPPA2 levels are correlated across time. If the levels are correlated, would it be possible to use this to further improve the classifier, i.e. using PAPPA2 levels at both 20 and 28 weeks to further improve accuracy?

Minor points:

I am not a big fan of venn diagrams with more than three circles. I prefer upset plots and I would recommend the authors to use them for greater clarity.

On line 430 it says that each case was paired with four unique controls and that no control was shared between cases. However, on line 439 it says that 25 term cases were matched to 96 controls? How does this work out when $25 \times 4 = 100$?

(Remarks on code availability)

The documentation is very nice and comprehensive.

Version 1:

Reviewer comments:

Reviewer #1

(Remarks to the Author)

REVIEW OF REVISED MANUSCRIPT
RESEARCH SUMMARY:

[UNCHANGED]

In their manuscript Gong et al report on the use of cfRNA as present in maternal plasma to predict pregnancies complicated by the combination of Preeclampsia and Fetal Growth Restriction ([PE+FGR]). Using biobanked maternal plasma samples prospectively collected in the POP study, the authors conducted a 1:4 case:control study comprising longitudinally collected samples (up to 4 sampling timepoints across gestation) of 39 women developing [PE+FGR] in their pregnancy and 156 control pregnancies, i.e., women who didn't experience obstetric complications during pregnancy. The study population was split into a discovery and (internal) validation cohort, whereby the cases (n=15) in the discovery cohort consisted of women requiring delivery for PE+FGR before 37 weeks of gestation (preterm delivery), and the cases (n=24) in the (internal) validation cohort consisted of women requiring delivery for PE+FGR from 37 weeks onward (term delivery).

Based on the levels of cfRNA in patient samples collected at ~ 28 weeks of gestation, the authors selected 17 genes of interest based on their ability to differentiate cases from controls, i.e, the predictors. Following an exploratory evaluation of 11 machine learning methods based [PE + FGR] prediction model performance, elastic net was used to develop final prediction models with n=2 to n=10 predictors in the 28 weeks discovery data.

[PE+FGR] prediction performance (AUC) of the final models was observed in the earlier gestational age windows (12, 20 wks) in the discovery cohort and in the 4 gestational age data sets (12,20,28,36 wks) of the internal validation cohort.

Prediction performance increased across the gestational age data sets, from virtually non-predictive at 12 weeks to moderately predictive or diagnostic at 28 weeks and 36 weeks. It was found that the 2 predictor model based on the cfRNA levels for PAPP2 and LEP were most performant as compared to the models with more predictors (n>2). Interestingly, the individual predictors, PAPP2 and LEP can have equivalent stand-alone prediction compared to the model; this is true for PAPP2 at all gestational age windows and LEP at 36 weeks.

Finally, the authors applied the [PAPP2 + LEP] prediction model to the cfRNA data from Munchel et al (1). This study involved samples taken at ca. 30.5 wks of gestation from 40 early onset preeclampsia cases (preeclampsia diagnosis before 34 week of gestation) with (45%) or without (55%) FGR and 73 control pregnancies. Remarkably, PAPP2, LEP and the [PAPP2 + LEP] prediction model were highly diagnostic in the Munchel data set with AUCs of ca. 0.95.

REVIEW OF THE AUTHOR ANSWERS TO PEER REVIEW #1

1.1 Overall, the authors addressed the comments regarding validity adequately in their revised manuscript; the reviewer notes that the PAPP2 findings of Gong et al are in good agreement with the recent report of Elovitz et al <https://doi.org/10.1038/s41467-025-58157-y> ; adding further clout to the research from Gong et al.

1.2 OK; apply the following editorial corrections to lines 86-90: For the exonic enrichment we use customised probes that were tailored to the placenta transcriptome and sequenced an average of 300M reads per sample. We evaluate 11 machine learning methods for selecting the best combination of predictors. Finally, we validate the predictive performance of our models using an externally-sourced RNA-seq datasets where the cohort and the sequencing data were generated independently.

Update the Table in Figure 1 Panel B with adaboost and remove ENet2; apply same correction in github; Figure 8.3 in github is also not updated.

1.3 OK; It is suggested to modify the language in lines 385-390 as follows: Notably though, the cfRNA model yielded an AUC of 0.95 in a geographically and clinically divergent cohort from the study used to generate the model; the equivalent protein data was not available for comparison.

1.4 OK; Reviewer #1 agrees that with PAPP2 + LEP the authors have identified and confirmed a strong panel to predict (preterm) [PE + FGR]. At the same time Reviewer #1 also notes that the additional data as generated by Gong whereby the models were trained to predict [all PE + FGR] as well, are indicative for the existence of models whereby PAPP2 + LEP can be differentially be supplemented with other cfRNAs for improved prediction of either preterm [PE + FGR] or term [PE + FGR]. The authors may want to explore this further in another paper.

1.5 To maintain the high level quality throughout, it is suggested that the authors conduct a thorough final cross check of the github content against the main manuscript upon consolidation of the latter, as not all changes to the main manuscript are updated in the github

1.6 OK

1.7 OK, the authors are thanked to provide this additional information

1.8 OK, please apply the same changes in the github

1.9 OK. The data in the new supplementary #7 are very effective; the authors are commended to generate this informative summary Data

1.10 OK, as noted earlier. The authors should ensure that all figures and tables in the main manuscript as well as in the supplementary materials (github) correctly reflect the addition of adaboost.

1.11 OK, very instructive – thanks for generating,

1.12 OK, the authors are commended for adding this significant piece of modelling to their work; the authors may want to pursue in separate research how the average longitudinal profiles look for the [preterm PE + FGR] and [term PE + FGR] patients;

1.13 Ok; thank you. Reviewer #1 appreciates the authors performed the all PE + FGR prediction modelling. As mentioned in 1.4, review of the additional data as presented in supplementary table 9 and supplementary Figure 9.2 indicate that there is scope for modulating prediction performance in function of presentation (preterm vs term). The comments regarding the difference in ENET performance is superfluous (lines 288-290), observing that the same 2-marker panel was selected is sufficient; the performances are virtually identical and well within each other's 95% CI intervals.

1.14 OK.

1.15 OK.

1.16 OK.

1.17 OK; differential changes in function of GA ("slopes") may still be present; the authors may want to investigate this in separate research.

1.18 OK, update Figure labels accordingly in Figures 3 and 4

1.19 OK; re. language in lines 144, 149, 172: the reviewer assumes that Gong et al is applying the Benjamini-hochberg FDR control method controlling FDR at $\alpha = 0.05$ and calculating the Benjamini-Hochberg adjusted p-values. It is noted that these p-values are not really estimates of the probability. The better option would be to evaluating each individual P value to its Benjamini-Hochberg critical value.

1.20 OK

1.21 OK

RESEARCH REVIEW

A) VALIDITY OF THE RESEARCH: The manuscript & supplementary website <https://obsgynaecam.github.io/cell-free-rna-2024/> of Gong et al covers in adequate detail the research conducted.

B) SIGNIFICANCE OF THE RESEARCH: The manuscript has clarity in its objectives. At least three objectives are identifiable to the reviewer

a. Demonstration that the authors cfRNA analytical and data analysis work flow is robust.

b. Demonstration that circulation PAPP2- and/or LEP- cfRNA levels have clinical application potential in monitoring the development of the pregnancy complication PE + FGR

c. Independent validation of PAPP2- and LEP- cfRNA results as biomarkers for preeclampsia as reported by others

C) DATA AND METHODOLOGY: In general Gong et al provided high level detail in terms of study design, sample work up, cfRNA analysis, data processing, statistical analysis and model development. The quality of the dedicated website materials (<https://obsgynaecam.github.io/cell-free-rna-2024/>) is remarkable, yet needs some re-checking for consistency with manuscript post addressing peer review comments.

D) ANALYTICAL APPROACH:

a. The practical experimental design is sufficiently elaborated

b. In addressing the peer review comments Gong et al provided sufficient rationale for their multivariable model development approach.

E) CLARITY AND CONTEXT:

a. The authors should carefully re-read the revised manuscript and firm up their language at places; there are some missing articles, some minor spelling mistakes in the changes applied etc.

b. Carefully check that changes made in the research following peer review are carried through in the figures (main article / supplementary / github) as well as in the materials beyond the main manuscript (supplementary / github)

c. The graphs in Supplementary Figures 1A and 1B did not render content in the pdfs provided; they are available at the github; please ensure the supplementary info is complete.

(Remarks on code availability)

Reviewer #2

(Remarks to the Author)

The authors have addressed all of my comments.

(Remarks on code availability)

Response to reviewers' comments

NCOMMS-24-79680

Reviewer 1

#	Comment	Response
1.1	VALIDITY OF THE RESEARCH: In general terms the manuscript & supplementary website https://obsgynaecam.github.io/cell-free-rna-2024/ of Gong et al covers in adequate detail the research conducted. The authors should consider improving transparency about the fact that essentially only the cfRNA data from the 28 weeks discovery samples were considered for model development and no longitudinal modelling was performed.	
1.2	SIGNIFICANCE OF THE RESEARCH: The manuscript lack clarity in its objective. At least three objectives are identifiable to the reviewer a) Demonstration that the authors cfRNA analytical and data analysis work flow is more robust/more fit-for-purpose than any of the recent cfRNA publications in the same space >> as presented there is no convincing argumentation provided that the key technical changes, i.e, study design, use of latest NGS (illumina) tools, sequencing depth, the selection of a particular cfRNA quantification method, the application of the discrete super learner approach to select a machine learning method, are more appropriate than workflows reported by others for the same / similar clinical applications (1–4). For instance, it is unclear why the authors did not consider AdaBoost as used by Munchel et al in their super learner approach.	We have added the following text to the end of the introduction to explain some of the unique aspects of our study. “We use samples from a well-phenotyped prospective cohort of 4512 pregnant women from which samples and data were collected at 4 time points in pregnancy. This cohort has been previously described (17, 18). (New lines 79-81) and “For the exonic enrichment we use customised probes that were tailored to the placenta transcriptome and sequenced average of 300M reads per sample. We evaluate 11 machine learning model for selecting the best combination of predictors.” (New lines 86-88). We have now included Adaboost in our evaluation and added this additional analysis to Table 2 and Figure 3A.

1.3	b) Demonstration that circulation PAPP2- and/or LEP- cfRNA levels have clinical application potential in monitoring the development or pregnancy complications >> Justification of the selection of PE+FGR as the clinical application. The introduction appears to set up the narrative that the clinical application is about identifying cfRNA biomarkers for placental insufficiency, i.e., the intersect of FGR and (early onset) PE. Monitoring pregnancy for placental insufficiency would have clinical relevance. However, clarity of the intended clinical application is not maintained throughout. If the authors would to highlight clinical application potential, benchmarking PAPP2 and LEP cfRNA prediction against circulating s-Flt1 and PIGF will be instructive; the POP study likely has this data readily available (5,6).	We selected the PE with FGR as the clinical outcome for study because this is the strongest phenotype. We have added the following text to clarify this; “We selected women with pregnancies complicated by both PE with FGR as this combination is a severe phenotype and likely to have the strongest signal in a cfRNA data set.” (New lines 81-83). We have added the following text to the discussion comparing the PAPP2 and LEP cfRNA data with the sFLT1:PIGF ratio. “The sFLT1:PIGF ratio in maternal serum is used as a clinical diagnostic marker for PE. We have assessed this in the POPS cohort and the AUC was 0.95 (95% CI=0.90-1.0 at 28 weeks gestation and 0.92 – 0.99 at 36 weeks gestation) (35). Both the cfRNA and the protein tests were strongly associated with PE with FGR but the protein test appeared to perform better within sample (ie without testing in a separate validation cohort). In contrast, the cfRNA model yielded an AUC of 0.95 in a geographically and clinically divergent cohort from the study used to generate the model. (New lines 385-390)
1.4	c) Independent validation of PAPP2- and LEP- cfRNA results as biomarkers for preeclampsia as reported by others >> In view of the variability in analytical work ups and data analysis pipelines, the work of Gong et al. builds the body of scientific evidence for these markers; this is certainly a significant observation. Identification and clear messaging of what the authors consider the main scientific contribution of their work adds will improve impact of their manuscript.	We appreciate that the reviewer considers that our manuscript is “certainly a significant contribution”. We have added the following text to end of the discussion to provide clearer messaging as requested: “This combination of features in the experimental design is in contrast to other studies as is our finding that quantitation of only 2 mRNAs is sufficient to predict PE with FGR. This may indicate that with the appropriate design a simple model is sufficient to make meaningful prediction.” (New lines 464-466)
1.5	DATA AND METHODOLOGY: In general Gong et al provided high level detail in terms of study design, sample work up, cfRNA analysis, data processing, statistical analysis and model development. The quality of the dedicated website materials	We appreciate the very complementary comments about the high quality of our code repository.

	(https://obsgynaecam.github.io/cell-free-rna-2024/) is remarkable. Whereas the code was not tested / verified, review of the within code annotation suggests that the code was developed by (a) skilled scientist(s).	
1.6	Some sporadic misspellings were observed: f.i., gestion instead of gestation, shrink instead of shrink,...	We have corrected both of these errors.
1.7	The manuscript can be improved by addressing the following items: Salmon is selected as the preferred RNA-seq quantification method based on a thorough evaluation of correctly calling fetal sex based on chromosomal Y transcripts. Yet, it remains unexplained why the correct calling of fetal sex is indicative for Salmon being the better method for comprehensively identifying / quantifying cfRNA over the other methods.	The Y chromosome harbours multiple copies of Y-chromosome gene family members and many of these have homologues on the X-chromosome. This makes alignment (and quantitation) challenging. In addition, the Y-chromosome transcripts are lowly expressed in plasma. Therefore, a quantitation method that is able to reliably call fetal sex based on the detection of Y-chromosome encoded transcripts would deal effectively with these technical challenges. Pragmatically, Y-chromosome based fetal sex determination is a tractable metric on which to evaluate quantitative performance and has been used by others [Godfey 2020, citation #54]. We did describe this in the Supplementary information but have added some text to expand on our reasoning.
1.8	Selection of the so-called core genes is based on an overlap of several ranking exercises. Without rationale which eg refers to (different) underlying statistical assumptions used in DESeq2, edgeR and the AUC and AIC statistics as derived from the same univariable logistic regressions, this selection appears largely arbitrary and doesn't justify labelling the 17 genes as "core genes" as this may infer they are "core" to development of placental insufficiency; without justification of the latter, less "leading" terminology should be adopted.	We have reworded this throughout the manuscript to read "shared"
1.9	When it comes to the use of the discrete super learner approach: The ranking criteria (Rank Sum and Mean Rank) remain unexplained, albeit they can be inferred from the results in supplementary data #6. Based on the statistics	We have added the following text to explain the ranking criteria, "Finally, we ranked each method and identified the best method, i.e. the one with the lowest sum of the ranks across 2 to 6 cfRNAs in the model". (New lines 200-201).

	presented in Fig 3, panel A, it is not apparent to the reader whether there is a clear-cut best ML, especially when one accounts for the fact that the final (robust) model put forward is a simple 2 predictor model. Naming the genes selected in the different models in supplementary data #6 in the same fashion as supplementary data #7 would be helpful to the reader to assess to what extent the different ML select the same predictors. Based on the data presented, the application of the discrete super learner approach appears of limited added value in this study; the authors may want to argue otherwise.	Supplementary Data #6 (now #7) is the summary of the 5-fold CV where the presented “Mean AUC” is the average across 25 (i.e. 5-fold x 5-repetition) AUCs tested on the held-out folds during the cross-validation. In contrast, Supplementary Data #7 (now #8) presents the AUC from the training model using the entire dataset (i.e. no fold) where the cfRNAs were selected by the best method (i.e. Elastic net) chosen from the cross-validation. Therefore, it is not appropriate to provide the genes in Supplementary Data #6 (now #7) as this data do not provide the predictors in the final model. Nonetheless, we have added a new table (new Supplementary Dataset #6) to show the list of cfRNAs identified by the various models during the cross-validation setting. This dataset serves the source of Supplementary Data #6 (now #7), so we hope it provides sufficient detail. The reviewer is correct that most of the methods evaluated in the discrete super learner do perform similarly. However, without carrying out this analysis we would not have known this. Comparison of analytical methods as we describe (even if they turn out to have similar performance) justifies the final selection.
1.10	As mentioned earlier, explanation of the non-consideration of adaboost as used by Munchel et al would also be helpful (e.g. only using non-ensemble ML techniques?).	As described in 1.2 above, we have now evaluated AdaBoost and added this analysis to Table 2 and Fig3A.
1.11	Given the simplicity of the final model, it would be helpful to provide plots (in supplementary) of the z-scores for PAPP2 and LEP with case and control labels in a simple x-y scatter for each of the predictions in Figure 4 – Panel D.	We have created the xy-scatter plot as requested in Supplementary Information Figure 3.
1.12	ANALYTICAL APPROACH: The practical experimental design as outlined in the Sample collection paragraph is an ideal preparation for establishing longitudinal inpatient data as for each patient all gestational timepoints are processed in the same batch. This begs the question why no longitudinal data analysis / modelling was planned/performed by the authors; please comment.	We have now carried out a longitudinal analysis and added this data – New Fig 5B and Supplementary Information Figures 4, 6 and 6 and Supplementary data 10,11 and 12. The text has been added: “Longitudinal analysis of cfRNAs in healthy pregnancy Having identified gestational age-related changes in circulating LEP and PAPP2 cfRNAs (Figure 5A), we sought to identify all cfRNAs that change in abundance as gestation progresses. For this analysis, we used a total of 96 healthy samples from the validation cohort (i.e. term delivery) and carried out two types of analyses: 1) to identify cfRNAs that differ between specific gestational ages: (i.e. 12wk to 20wk, 20wk to 28wk, and 28wk to 36wk), and 2) a longitudinal study of cfRNAs across the four gestational ages. We found a total of 129 genes for which the cfRNA levels changed significantly (76 from 12wk to 20wk; 64 from 20wk to 28wk; 55 from 28wk to 36wk, see Supplementary Data 10). LEP and PAPP2 were among these 129

		genes, where LEP cfRNA decreased from 12wk to 20wk (adjusted p-value=1.2×10^{-30}, Benjamini-Hochberg method) and PAPP2 cfRNA increased between 20wk to 28wk (adjusted p-value=6.9×10^{-5}) and 28wk to 36wk (adjusted p-value=2.6×10^{-39}). Of the 129 genes, 16 (ALPP, C2orf72, CAPN6, CSHL1, CYP11A1, HSD3B1, LINC00967, MUC15, PAPP2, RAB3B, SVEP1, TEAD3, TFAP2A, TINCR, TMEM54, and VGLL3) significantly increased between all intervals (Figure 5B). For the longitudinal analysis, we considered a random intercept model (i.e. allowing different baseline levels for each participant) with both a linear and quadratic term of gestational age as fixed effects (see Methods). When the fitness of mixed effect model was ranked by the coefficient of determination (i.e. r^2), CSHL1 showed the highest marginal r^2 (0.689), followed by CAPN6 (0.688) and RAB3B (0.631) – see Supplementary Data 11. All the 16 aforementioned genes were highly ranked – 10 of them (CSHL1, CAPN6, RAB3B, SVEP1, VGLL3, ALPP, LINC00967, PAPP2, HSD3B1, and MUC15) were within the top 12 (Supplementary Data 11 and Supplementary Figure 4). When the models were assessed by the significance of fixed effects, LEP was the most significant (adjusted p-value=9.7×10^{-91}, Benjamini-Hochberg method), followed by CSHL1 (adjusted p-value=2.5×10^{-44}) and PAPP2 (adjusted p-value=4.53×10^{-32}) (Supplementary Figure 5). Indeed, LEP showed the highest absolute coefficient values of fixed effects when the gestational age was considered as its linear (-5.62) and quadratic (1.08) term in the mixed effect model. Finally, when only the linear term was considered in the model (i.e. no quadratic term), TTPAL was identified as the most significantly decreasing cfRNA (adjusted p-value=8.5×10^{-59}) by the gestational age, followed by KIAA1147 (adjusted p-value=2.7×10^{-50}) and RPL37AP1 (adjusted p-value=5.4×10^{-41}) (Supplementary Data 12 and Supplementary Figure 6).” (New lines 341-369)
1.13	Development of robust multivariable predictors is constrained by the number of cases in the discovery cohort. Given the low number of effective cases (n=13 at 28 weeks, see tabulation of discovery cohort on https://obsgynaecam.github.io/cell-free-rna-2024/sample.html), the choice for a final 2 predictor model is appropriate. The authors may want to consider re-analysing the data without splitting the data set and use the complete POP (28 weeks) sample set for model development as advised (7). This may lead to a robust n>2 predictor model with better performance and side	As this reviewer has requested we re-analysed the data without splitting the datasets and used all the 28wkGA samples by combining both discovery (preterm) and validation (term) cohort. We developed 2- to 10-cfRNA models from this combined dataset and compared their predictive performances to the original (preterm) models. For 2-cfRNA models, LEP and PAPP2 were also selected by Elastic net in the combined dataset, but when validated in Munchel dataset, its predictive performance was slightly lower (AUC=0.944, 95% CI=0.901-0.988) than that of the original model (AUC=0.951, 95% CI=0.912-0.990) trained in discovery cohort only. For >2-cfRNA models, cfRNAs were selected differently in the combined dataset and their predictive performances appeared to be improved for 6-cfRNAs or more. To note, LEP and PAPP2 were always selected in the 2- to 10-cfRNA models. We have created Supplementary Data 9 and Supplementary Figure 2 for detailed descriptions.

	step the risk that the discovery cohort and the internal validation cohort represent possibly different disease subtypes, as early onset presentations of placental insufficiency (<34 weeks of gestation) is oftentimes considered to be distinct to later presentations of placental insufficiency. The applied study population split may therefore be biased to select for early onset disease markers. It is noted that the Munchel external validation cohort is an early onset PE +/- FGR cohort, which may explain why the PAPP2 + LEP model has better performance in Munchel as compared to the internal validation cohort. It is observed that the authors did not explicitly mention this feature of the Munchel study population.	We have added the following text to the manuscript: “We evaluated models generated using all the 28wkGA samples from both discovery (preterm delivery) and validation cohort (term delivery) cohorts. The same two-cfRNAs (LEP+PAPP2) were chosen by Elastic net in this combined model. However, the predictive performance of this model was slightly lower in the external Munchel dataset (AUC=0.944, 95% CI=0.901-0.988) than that of the original model trained in discovery cohort only (Supplementary Data 9 and Supplementary Figure 2).” (New lines 288-293).
1.14	It is noted that the 2 predictors selected for the model are the 2 only predictors (2/17) which are upregulated in the PE+FGR cases (table 1). Is this to be expected or is there a possible predictor selection bias in the ML chosen. Please comment.	As we have evaluated multiple ML methods and used 5-fold cross-validation it is very unlikely that all 11 methods would suffer the same bias.
1.15	CLARITY AND CONTEXT: The transparency of the manuscript would benefit from clarifying the following Selection of PE+FGR as the clinical application. The introduction appears to set up the narrative that the clinical application is about identifying cfRNA biomarkers for placental insufficiency, i.e., the intersect of FGR and (early onset) PE. This clarity is not maintained throughout.	We have been careful through the manuscript to talk about “PE combined with FGR.” To ensure this remains clear throughout we have added “and we have demonstrated this for the combination of PE with FGR.” To the end of the discussion (New line 510-511)
1.16	There is conflation of PE and FGR in terms of maternal presentation in lines 50 – 54 ; endothelial dysfunction is considered a hallmark of PE and not FGR.	We have clarified this by specifying that the endothelial dysfunction is in PE (New line 54).
1.17	The discovery cohort is effectively a 28 weeks only cohort: DEGs were only selected from this timepoint; model development only considered	We agree that the cfRNA varies at different gestational ages. We have added a much fuller description of this in response to point 1.12 above. We also agree that making a prediction earlier in pregnancy could be useful. Thus, we sought to identify cfRNAs

	this timepoint. The longitudinal nature of the data was not leveraged for model development. Performance of the 28 weeks model is solely evaluated at the different timepoints. The lack of performance in the earlier gestational ages may indicate that the composition of the cfRNA repertoire at 28 weeks is different than these at earlier timepoints, making the model ill-suited for risk prediction in the first half of pregnancy. Longitudinal analysis of cfRNA / gene expression may have elicited true monitoring.	that differ between the cases and controls at 12 wkGA and 20wkGA. However, there were no DEGs at either timepoint.
1.18	Add a brief clarification that only/mainly the mRNA fraction of cfRNA is being explored, this would improve readability as the manuscript switches regularly between cfRNA and mRNA terminology	We have edited the text and now consistently use “cfRNA” to avoid possible confusion.
1.19	The Benjamini and Hochberg correction is better characterised as a method to control false discovery rate than a multiple-test correction	We have corrected this (New line 647)
1.20	The authors benchmark their data to literature investigating use of cfRNA for prediction/diagnosis of preeclampsia. No attempt was made to benchmark to FGR literature; cf. ref 14 in gong et al.	To the best of our knowledge there is no published work describing cfRNA analysis in cases of FGR. Thus, there is nothing to compare with. The reference the reviewer appears to be pointing to is Gong et al (PMID 29997303) which describes maternal serum metabolites in cases of FGR and RNAseq from healthy male and female placentas.
1.21	In absence of data on PE without FGR and FGR without PE, the question remains what pregnancy complication the reported cfRNA signatures are specifically reflecting. The authors are invited to briefly comment on this in the Discussion. See also earlier comments on clarity of clinical application.	We have specified that the analysis is restricted to cases of PE with FGR and added text to the discussion to reflect this. (New lines 507 and 510)
1.22	Remarks on code availability: The code as available at the provided supplementary website (https://obsgynaecam.github.io/cell-free-rna-2024/) was visually inspected; the within code annotation was comprehensive. No attempts were made to install and run code.	Thank you

	The quality of information provided at supplementary website was high.	
--	--	--

Reviewer 2

#	Comment	Response
	Given the large number of ML models explored, it is clear that the signal is very robust. The generalization to the Munchel et al study is also reassuring. Moreover, the authors deserve credit for the Github repository containing the code to reproduce the results. I wish that more authors put in a similar effort for documenting their work. Nevertheless, I believe that the following major concerns must be addressed:	Thank you, we appreciate these complementary remarks.
2.1	I do not understand the point of fig 2a? Why is it meaningful to test the distribution of uncorrected p-values? In my opinion, there is little value in examining the distribution of p-values. As far as I am concerned, there is no interesting or meaningful model stipulating how they should be distributed, so I do not see the point of this analysis or test.	A pronounced skewing of the distribution of the uncorrected p-values, with an excess of low values (as we show in fig 2) is a simple way to determine whether there is any meaningful signal in the data. If the distribution is flat, with no excess of low p-values then even if there are some corrected p-values below 0.05. This type of plot was used in the development of a Bayesian gene-array analysis package (Cyber-T, Hung, Baldi & Hatfield 2001, PMID 12130640).
2.2	I am curious about the glParallel model with two genes in fig 3A. I understand that overall this model performs poorly. However, given that the authors later focus on models with two genes, I am wondering if this model could be useful after all. I encourage the authors to investigate, in case it turns out	As suggested, we further investigated 2-predictor models where glParallel was superior to any other method we compared based on the cross-validation– we have added more detailed results of 5-fold cross-validation (along with 3-6 predictor models) in Supplementary Dataset 6. When the model was trained on the entire dataset (i.e. 28wk discovery cohort), glParallel identified HBQ1+PAPPA2 as the final model with AUC of 0.921 (95%CI=0.835-1.000) – it was higher than that of ENet (LEP+PAPPA2; AUC=0.903). However, when its predictive performance was tested internally (validation cohort) and externally (Munchel), the validated AUCs from glParallel were not as good as those from ENet:

	that they have missed a high-performing model.	    Dataset     Validation Munchel   Method Predictor 12wk 20wk 28wk 36wk      glParallel HBQ1+PAPPA2 0.517 0.635 0.689 0.698 0.873   ENet LEP+PAPPA2 0.556 0.695 0.800 0.807 0.951    This suggests that considering multiple number of predictors in the models (2-6 predictors in current study) during the cross-validation stage was indeed a right approach, because we would have selected glParallel as the best method if only a 2-predictor model were considered. In addition, this is consistent with our finding that the validated AUCs gradually deteriorated when the number of cfRNAs increased in the predictive models, suggesting over-fitting (Figure 4A).			Dataset							Validation				Munchel	Method	Predictor	12wk	20wk	28wk	36wk		glParallel	HBQ1+PAPPA2	0.517	0.635	0.689	0.698	0.873	ENet	LEP+PAPPA2	0.556	0.695	0.800	0.807	0.951
		Dataset																																			
		Validation				Munchel																															
Method	Predictor	12wk	20wk	28wk	36wk																																
glParallel	HBQ1+PAPPA2	0.517	0.635	0.689	0.698	0.873																															
ENet	LEP+PAPPA2	0.556	0.695	0.800	0.807	0.951																															
2.3	How about using partial AUC (pAUC) instead of AUC for ranking the models? In my opinion, AUC may put too much emphasis on regions of the ROC curve that are not relevant in practice. For example, if the intention is to run the pregnancy test with high specificity (I do not know if this is the case since I am not a clinician), then using a pAUC with a cut-off at for example 0.2 on the x-axis might be more appropriate. This could potentially result in some of the method rankings changing.	We appreciate this reviewer's proposal using partial AUC instead of AUC. It could be useful in certain clinical settings where a controlled range of sensitivity and specificity is desired. However, we do have such a prior information as the use of cfRNAs in antenatal diagnostics is new and this area is not well studied. Therefore, we decided to be more conservative as we are not sure how much pAUC could be beneficial compared to AUC which is well-established and widely-used in the cross-validation for a binary classification. By presenting the actual AUC plots, the interested reader will be able to make assessments of performance across the range of specificity.																																			
2.4	If I understand correctly, for most participants in the study there is cfRNA from more than one time point. However, this information does not appear to be used at all in the analysis. I would be interested for example for fig 5 to learn if the LEP and PAPPA2 levels are correlated across time. If the levels are correlated, would it be possible to use this to further improve the	We have added a systematic longitudinal analysis (see response 1.12 above) to address the question of gestational age-related changes. The LEP cfRNA shows a U-shaped curve over gestation where as PAPPA2 increases steadily. This is more clearly shown in new Supplementary Figure 5 where the fitted quadratic curves for LEP and PAPPA2 differ. In the discovery cohort, the Pearson's correlation coefficient between LEP and PAPPA2 cfRNAs was higher at 28wk ($\rho = 0.63$) than that at 20wk ($\rho = 0.56$). In the validation cohort, the correlations were even weaker (0.59 and 0.32 at 28wk and 20wk respectively). We have added this information to the manuscript. (New lines 341-369)																																			

	classifier, i.e. using PAPP2 levels at both 20 and 28 weeks to further improve accuracy?	Pragmatically, a predictive test that only requires a single blood sample is much easier to implement than one that requires serial samples.
--	--	--

Minor points

2.5	I am not a big fan of venn diagrams with more than three circles. I prefer upset plots and I would recommend the authors to use them for greater clarity.	We also appreciate the clarity of the upset plot, however because space is limited in this figure we would prefer to retain the current Venn diagram.
2.6	On line 430 it says that each case was paired with four unique controls and that no control was shared between cases. However, on line 439 it says that 25 term cases were matched to 96 controls? How does this work out when $25 \times 4 = 100$?	Apologies, this was a typo and the number is 24 (as already stated in Supplemental Table 1 and in the "Overview of the current study" in the Results section). Therefore, the number of controls ($n=96$) is still valid, i.e. $24 \times 4 = 96$. We have corrected this error. (Now line 500)
2.7	Remarks on code availability: The documentation is very nice and comprehensive.	Thank you